# On the Convergence of Two-Layer Kolmogorov-Arnold Networks with First-Layer Training

**Seyed Mohammad Eshtehardian**
Department of Electrical Engineering
Sharif University of Technology
Tehran, Iran
mohammad.eshtehardian@sharif.edu

**Mohammad Hossein Yassaee**
Department of Electrical Engineering
Sharif University of Technology
Tehran, Iran
yassaee@sharif.edu

**Babak Khalaj**
Department of Electrical Engineering
Sharif University of Technology
Tehran, Iran
khalaj@sharif.edu

## Abstract

Kolmogorov-Arnold Networks (KANs) have emerged as a promising alternative to traditional neural networks, offering enhanced interpretability based on the Kolmogorov-Arnold representation theorem. While their empirical success is growing, a theoretical understanding of their training dynamics remains nascent. This paper investigates the optimization of a two-layer KAN in the overparameterized regime, focusing on a simplified yet insightful setting where only the first-layer coefficients are trained via gradient descent.

Our main result establishes that, provided the network is sufficiently wide, this training method is guaranteed to converge to a global minimum and achieve zero training error. Furthermore, we derive a novel, fine-grained convergence rate that explicitly connects the optimization speed to the structure of the data labels through the eigenspectrum of the KAN Tangent Kernel (KAN-TK). Our analysis reveals a key advantage of this architecture: guaranteed convergence is achieved with a hidden layer width of $m = \mathcal{O}(n^2)$, a significant polynomial improvement over the $m = \mathcal{O}(n^6)$ requirement for classic two-layer neural networks using ReLU activation functions and analyzed within the same Tangent Kernel framework. We validate our theoretical findings with numerical experiments that corroborate our predictions on convergence speed and the impact of label structure.

## 1 Introduction

Neural networks have become the cornerstone of modern machine learning. Despite their success, the nested composition of linear transformations and fixed nonlinearities (e.g., ReLU) creates an opaque structural complexity, often relegating these models to black boxes. This opacity makes it difficult to interpret their decision-making processes, posing a significant barrier in high-stakes domains where trust and transparency are paramount. Kolmogorov–Arnold Networks (KANs) Liu et al. (2025) offer a fundamentally different approach, with an architecture inspired by the Kolmogorov–Arnold representation theorem (Kolmogorov, 1961; Braun & Griebel, 2009). This theorem establishes that any continuous multivariate function can be decomposed into a nested sum of univariate functions, which are far easier to interpret.

Although the idea of building networks upon this theorem is not new, early attempts based directly on its two-layer structure struggled due to the potentially non-smooth and complex nature of the inner functions, making them difficult to learn in practice (Sprecher & Draghici, 2002; Köppen, 2002; Lin & Unbehauen, 1993; Lai & Shen, 2021; Leni et al., 2013; Fakhoury et al., 2022). The key innovation

of modern KANs was to extend this shallow structure into a deep, multi-layer architecture, analogous to MLPs. This design mitigates earlier learning difficulties and shifts the paradigm: whereas MLPs place fixed nonlinearities at nodes, KANs place learnable univariate activation functions on the edges. This architectural choice not only improves interpretability but also enhances parameter efficiency. These learnable edge functions are typically parameterized as linear combinations of basis functions, such as B-splines (de Boor, 2001; Schumaker, 2007). More recent approaches have expanded this idea using alternative basis families, including Rational Polynomials Aghaei (2024b), Chebyshev Polynomials SS et al. (2024), and Radial Basis Functions (RBFs) Li (2024). In addition, recent works such as Delis (2024); Hu et al. (2025); Zhao et al. (2025); Bozorgasl & Chen (2024); Seydi (2024); Aghaei (2025) have introduced new classes of basis functions, further broadening the expressive power and adaptability of KANs.

The rapid emergence of KANs has led to exploration across diverse application domains. In computer vision, KAN-based convolutional architectures have demonstrated superior performance compared to traditional CNNs Bodner et al. (2024); Drokin (2024), and have been successfully integrated into U-Net models for medical imaging (Li et al., 2025). For sequential data, Temporal KANs were introduced in Genet & Inzirillo (2024), where KANs replace the standard neural components in RNNs, yielding improved accuracy on complex time-series tasks (Han et al., 2024; Xu & Wang, 2024). KANs have also been applied in reinforcement learning, achieving higher accuracy and performance with significantly fewer parameters Guo & Liu (2024); Kich et al. (2024), as well as in time-series analysis tasks (Huang et al., 2025; Zhou et al., 2025). Similar performance gains have been reported in graph neural networks (Zhang & Zhang, 2024; Fang et al., 2025; GuoguoAi et al., 2025). Beyond these, KANs have shown strong potential in scientific machine learning, particularly for solving partial differential equations, where they outperform physics-informed neural networks (PINNs) (Wang et al., 2025b; Toscano et al., 2024; Aghaei, 2024a). The architecture has also been adapted for Transformers, showing promise for large language models (Yang & Wang, 2025). Furthermore, Yu et al. (2024) demonstrated that KANs outperform MLPs on datasets constructed from symbolic formulas. Comprehensive surveys and further results are available in (Ji et al., 2024; Rigas et al., 2024; Howard et al., 2024; Cheon, 2024; Qiu et al., 2024; Polar & Poluektov, 2021; Lee et al., 2025).

Alongside these empirical successes, a growing body of theoretical work has begun to establish a rigorous foundation for KANs. Several works have investigated the role of initialization, including interpolation-based, random-based, and hybrid schemes designed to reduce the computational cost of KAN initialization and ensure stable training across different basis functions (Rigas et al., 2025). On the expressiveness side, Wang et al. (2025a) showed that KANs are at least as expressive as MLPs and may exhibit reduced spectral bias. Generalization properties have also been studied Zhang & Zhou (2025), and other works explore deep learning alternatives to the classical Kolmogorov-Arnold representation theorem itself (Guilhoto & Perdikaris, 2025; Laczkovich, 2021).

On the optimization side, a wide range of algorithms have been proposed for training machine learning models Kingma & Ba (2015); Carmon et al. (2018), with convergence guarantees typically relying on smoothness, Lipschitzness, or convexity assumptions (Li & Orabona, 2019; Nesterov & Polyak, 2006; Duchi et al., 2011; Reddi et al., 2019; Ji & Telgarsky, 2019). For MLPs, Zhang et al. (2021) observed that gradient descent (GD) and stochastic gradient descent (SGD) often reach nearly global minima in practice, driving the mean squared error toward zero. However, understanding why simple gradient-based methods succeed in optimizing highly non-convex models such as MLPs and KANs remains a central challenge.

Substantial progress has been made in the overparameterized regime Du et al. (2019); Jacot et al. (2018); Arora et al. (2019); Chizat & Bach (2018a); Soudry & Carmon (2016); Soltanolkotabi (2017); Xie et al. (2017); Chizat & Bach (2018b); Soltanolkotabi et al. (2018); Vaswani et al. (2019); Oymak & Soltanolkotabi (2020); Allen-Zhu et al. (2019); Polaczyk & Cyranka (2023), where neural tangent kernel (NTK)–type analyses yield convergence guarantees for sufficiently wide networks. More recently, overparameterization requirements for two-layer networks have been sharpened: Polaczyk & Cyranka (2023) derive improved width bounds that ensure global convergence of GD through a refined analysis of the empirical Gram matrix. Extending this line of work to the KAN setting, Gao & Tan (2025) prove that a two-layer KAN converges to a global minimum when all parameters are jointly trained.

In this paper, we analyze the training dynamics of a two-layer KAN under a more constrained setting: only the first-layer coefficients are trained, while the second-layer coefficients are fixed after a random initialization. This setup, previously studied for standard neural networks Du et al. (2019); Arora et al. (2019), allows for a clearer analysis. Our contributions are as follows:

- We prove that for a two-layer KAN with only first-layer training, gradient descent converges to a global minimum, driving the training error to zero, provided the hidden layer is sufficiently wide.

- We derive a novel, label-dependent bound on the convergence rate, showing that the speed of convergence is determined by the projection of the label vector onto the eigenvectors of the corresponding KAN Tangent Kernel (KAN-TK).

- We show that the required width of the hidden layer for guaranteed convergence in our KAN setup is significantly smaller than that required for standard two-layer neural networks Du et al. (2019), highlighting a key parameter-efficiency advantage.

- We provide empirical evidence that corroborates our theoretical findings, demonstrating the faster convergence for wider networks and the impact of label structure.

## 2 PRELIMINARIES AND SETUP

### 2.1 KOLMOGOROV-ARNOLD NETWORKS (KANS)

A KAN's architecture is inspired by the Kolmogorov-Arnold representation theorem, which states that any continuous multivariate function $f : [0, 1]^d \to \mathbb{R}$ can be written as:

$$f(\boldsymbol{x}) = \sum_{q=1}^{2d+1} \Phi_q \left( \sum_{p=1}^{d} \phi_{p,q}(x_p) \right)$$

where $\Phi_q$ and $\phi_{p,q}$ are continuous univariate functions. While early attempts to build networks based on this theorem struggled Sprecher & Draghici (2002); Köppen (2002), the key innovation of modern KANs was to extend the two-layer structure of the theorem into a deep network, analogous to MLPs (Liu et al., 2025). In this architecture, learnable univariate functions, often parameterized as splines, are placed on the edges of the computation graph, while nodes simply perform summation. This is in stark contrast to MLPs, where linear transformations occur on the edges and fixed non-linear activations are applied at the nodes.

The learnable edge functions are typically represented as a linear combination of basis functions, $\phi(x) = \sum_i c_i B_i(x)$, where the coefficients $c_i$ are trainable parameters. A common choice for the basis functions $B_i(x)$ is B-splines, which are piecewise polynomials with favorable mathematical properties such as local support and controllable smoothness, making them well-suited for function approximation (Schoenberg & Whitney, 1953; de Boor, 2001; Schumaker, 2007). The original KAN architecture, for instance, uses cubic B-splines by default (Liu et al., 2025). To improve computational performance and explore different inductive biases, various alternatives have been proposed, including Radial Basis Functions (RBFs) Li (2024), Reflectional Switch Activation Functions (RSWAF) Delis (2024), Chebyshev Polynomials SS et al. (2024), Rational Polynomials Aghaei (2024b), and Fractional Jacobi basis functions (Aghaei, 2025).

### 2.2 THE TWO-LAYER KAN ARCHITECTURE

We focus on a two-layer KAN with a $d$-dimensional input $\boldsymbol{x}$, a hidden layer of width $m$, and a scalar output. The output $f(\boldsymbol{x})$ is defined as:

$$f(\boldsymbol{x}) = \frac{1}{\sqrt{m}} \sum_{p=1}^{m} \sum_{l=1}^{g} \beta_{pl} \phi_l(z_p) \quad \text{where} \quad z_p = \sum_{k=1}^{d} \sum_{j=1}^{g} \alpha_{pjk} \phi_j(x_k).$$

Here, $\{\phi_j\}_{j=1}^{g}$ are a set of $g$ basis functions (e.g., RBFs), $\alpha_{pjk}$ are the learnable coefficients for the first layer, and $\beta_{pl}$ are the coefficients for the second layer. The $\frac{1}{\sqrt{m}}$ factor is a standard scaling term used in overparameterization analysis (Jacot et al., 2018).

A schematic illustration of this two-layer KAN architecture is provided in Figure 1.

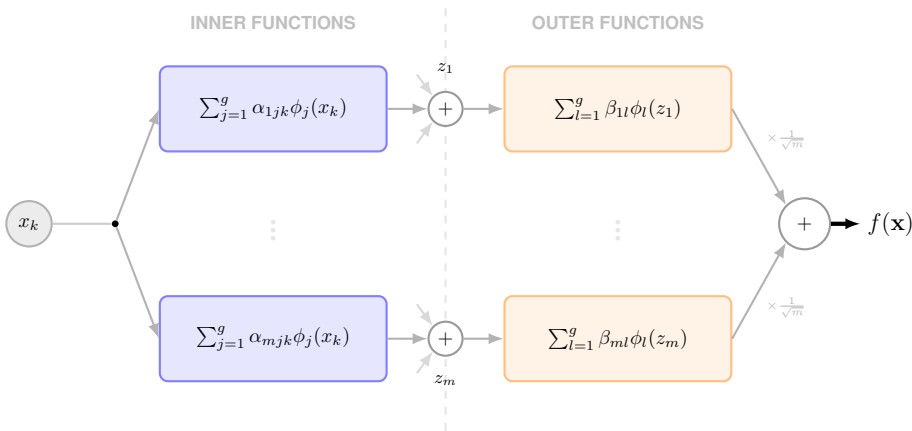

Figure 1: Architecture of a two-layer Kolmogorov-Arnold Network (KAN). The inner functions (blue) map inputs to intermediate latent variables $z$, which are then processed by outer functions (orange) and aggregated.

### 2.3 TRAINING DYNAMICS IN OVERPARAMETERIZED MODELS

Our analysis is situated in the overparameterized regime, where the number of model parameters far exceeds the number of training data points. In this regime, neural networks trained with gradient descent often exhibit a phenomenon known as "lazy training" Chizat & Bach (2018a), where the network weights remain close to their initial values throughout training. This allows the network's output to be well-approximated by a first-order Taylor expansion around its initialization.

This linearization gives rise to the Neural Tangent Kernel (NTK) Jacot et al. (2018), a deterministic kernel that governs the training dynamics of the network. For a two-layer MLP, it has been shown that if the network width is polynomially large in the number of data points $n$, gradient descent finds a global minimum, and the training dynamics are equivalent to kernel regression with the NTK (Du et al., 2019; Arora et al., 2019). Our work applies a similar analytical framework to the two-layer KAN architecture.

### 2.4 TRAINING SETUP AND PROBLEM FORMULATION

We analyze the network under the following training protocol:

1. **Initialization**: The first-layer coefficients $\alpha_{pjk}$ are initialized independently from a Gaussian distribution $\mathcal{N}(0, \sigma^2)$. The second-layer coefficients $\beta_{pl}$ are initialized independently and uniformly from the set $\{-1, +1\}$.

2. **Training**: Only the first-layer coefficients $\alpha = \{\alpha_{pjk}\}$ are updated using full-batch gradient descent. The second-layer coefficients $\beta = \{\beta_{pl}\}$ remain fixed throughout training (See Appendix E for more information).

Given a dataset $\{(\boldsymbol{x}_i, y_i)\}_{i=1}^n$, the goal is to minimize the mean squared error loss function:

$$\mathcal{L} = \frac{1}{2}\|\boldsymbol{y} - \boldsymbol{u}\|_2^2 = \frac{1}{2}\sum_{i=1}^n (y_i - f(\boldsymbol{x}_i))^2$$

where $\boldsymbol{u}$ is the vector of network outputs for all data points.

## 3 THE KAN TANGENT KERNEL

Our analysis relies on the concept of the *KAN Tangent Kernel (KAN-TK)*, which characterizes the training dynamics of our two-layer KAN in the infinite-width limit. For a general model $f_{\boldsymbol{\theta}}(\boldsymbol{x})$, the tangent kernel is defined as $H_{ij} = \langle \nabla_{\boldsymbol{\theta}} f_{\boldsymbol{\theta}}(\boldsymbol{x}_i), \nabla_{\boldsymbol{\theta}} f_{\boldsymbol{\theta}}(\boldsymbol{x}_j) \rangle$. In the lazy training regime, this kernel

remains nearly constant throughout training. Consequently, the complex, non-linear dynamics of the network can be accurately described by the much simpler process of kernel regression with this fixed kernel (Jacot et al., 2018). Additional explanations and details about tangent kernels are provided in Appendix A.1.

For our specific two-layer KAN with a 1D input and RBF basis functions, we can derive a closed-form expression for the KAN-TK in the infinite-width limit ($m \to \infty$). Since we only train the first-layer coefficients $\alpha$, the kernel is computed with respect to these parameters. In this section, we assume the basis functions $\phi_j(x)$ are Radial Basis Functions (RBFs), defined as:

$$\phi_j(x) = \exp\left(-\frac{(x-\mu_j)^2}{2\sigma^2}\right)$$

**Proposition 3.1** (KAN Tangent Kernel with RBF basis). *For a two-layer KAN with RBF basis functions and fixed second-layer coefficients, the tangent kernel with respect to the first-layer weights $\alpha$ in the infinite-width limit is given by $\boldsymbol{H}^\infty$. The entry $(\boldsymbol{H}^\infty)_{qr}$ (for $1 \leq q, r \leq n$) is:*

$$(\boldsymbol{H}^\infty)_{qr} = \sum_{j,l=1}^{g} \frac{\phi_j(x^q)\phi_j(x^r)\exp\left(-\frac{\mu_l^2}{\sigma^2}\right)}{\sigma^4} \left\{ \sum_{s,p} \phi_s(x^q)\phi_p(x^r)X_{psl}^{qr} + \mu_l^2 Z_l^{qr} + \sum_s b_s^{qr} Y_{sl}^{qr} \right\}$$

*where the auxiliary tensors are defined as follows:*

$$A_{kl}^{qr} = \phi_l(x^q)\phi_k(x^q) + \phi_l(x^r)\phi_k(x^r)$$
$$b_l^{qr} = -2(\phi_l(x^q) + \phi_l(x^r))$$
$$\boldsymbol{G}^{qr} = (\boldsymbol{I} + \frac{\boldsymbol{A}^{qr}}{\sigma^2})^{-1}$$
$$T_l^{qr} = \exp\left(\frac{\mu_l^2}{8\sigma^4}(\boldsymbol{b}^{qr})^T \boldsymbol{G}^{qr} \boldsymbol{b}^{qr}\right)$$
$$Z_l^{qr} = \sqrt{\det(\boldsymbol{G}^{qr})} T_l^{qr}$$
$$Y_{sl}^{qr} = -\frac{\mu_l^2}{2\sigma^2}\sqrt{\det(\boldsymbol{G}^{qr})}(\boldsymbol{G}^{qr}\boldsymbol{b}^{qr})_s T_l^{qr}$$
$$X_{psl}^{qr} = \sqrt{\det(\boldsymbol{G}^{qr})}(\boldsymbol{G}^{qr})_{sp} T_l^{qr}$$
$$+ \frac{\mu_l^2}{4\sigma^4}\det(\boldsymbol{G}^{qr})(\boldsymbol{G}^{qr}\boldsymbol{b}^{qr})_s(\boldsymbol{G}^{qr}\boldsymbol{b}^{qr})_p T_l^{qr}$$

The derivation of this kernel is provided in Appendix A.2. The expression is highly complex and computationally intensive, scaling polynomially with the number of samples $n$. This makes it impractical for direct use in large-scale applications but provides a powerful tool for our theoretical analysis. Despite this complexity, we can use the kernel to perform regression and empirically verify its expressive power. Moreover, in our experiments we relied on this proposition specifically because it provides access to the eigenvalues and eigenvectors of the KAN-TK, which are essential for analyzing label alignment and convergence behavior.

## 4 THEORETICAL ANALYSIS

In this section, we present our main theoretical results. We first prove that gradient descent on our two-layer KAN converges to a global minimum with zero training error. We then refine this result by deriving a label-dependent convergence rate. Our analysis relies on a few standard assumptions.

**Assumptions.** We assume the following conditions hold:

1. **Basis Functions**: The basis functions $\phi_l$ are bounded, $|\phi_l(x)| \leq 1$, twice differentiable with bounded first and second derivatives, $|\phi_l'(x)|, |\phi_l''(x)| \leq 1$, and satisfy $\phi_l(0) = 0$.
2. **Positive Definite Kernel**: The infinite-width KAN Tangent Kernel $\boldsymbol{H}^\infty$ is positive definite, meaning its minimum eigenvalue $\lambda_0$ is strictly positive ($\lambda_0 > 0$).

3. **Bounded Data**: The training data labels are bounded, $|y_i| \leq 1$ for all $i$.

The assumption of a *Positive Definite Kernel* is standard in the analysis of overparameterized neural networks (Du et al., 2019; Arora et al., 2019). In particular, Gao & Tan (2025) shows that this assumption holds for KANs equipped with appropriate polynomial basis functions. Their Lemma 1 states:

**Lemma 4.1** (Positive Definite Kernels). *Assume that the basis functions are polynomials of degree less than $g$ and the transformation functions are hyperbolic tangent or sigmoid. Then $\lambda_0 > 0$ holds when all training samples are distinct. If no transformation is used, $\lambda_0 > 0$ holds when the training samples are linearly independent in the $\tilde{g}$-degree polynomial space:*

$$\{x_{i,1}, x_{i,1}^2, \ldots, x_{i,1}^{\tilde{g}}, \ldots, x_{i,d}, \ldots, x_{i,d}^{\tilde{g}}\}_{i=1}^n$$

*where $\tilde{g} = (g-1)^2$.*

The transformation $\psi$ (e.g., tanh or sigmoid) ensures the first-layer outputs lie within the domain of the polynomial basis, so KAN variants using such nonlinearities satisfy the lemma when samples are distinct. In the no-transformation case ($\psi(z) = z$), the lemma only requires linear independence in the relevant polynomial space. Empirically, using FastKAN Li (2024), we observe strictly positive minimum eigenvalues of the infinite-width KAN-TK across several input distributions (e.g., $3.29 \times 10^{-4}$ for `linspace` on $[-1, 1]$), supporting this assumption in practice.

## 4.1 GLOBAL CONVERGENCE

We first establish that under sufficient overparameterization, the training loss converges to zero.

**Theorem 4.2** (Convergence to Global Minimum). *Suppose the hidden layer width $m$ is sufficiently large and the initialization variance $\sigma^2$ is sufficiently small, i.e.,*

$$m \gtrsim \max\left(\frac{d^2 g^6 n^2}{\lambda_0^2} \log\left(\frac{n}{\delta}\right), n\right), \qquad \sigma = \mathcal{O}\left(\frac{\delta}{\sqrt{mng^3 d}}\right).$$

*Then, with probability at least $1 - \mathcal{O}(\delta)$ over the random initialization, the gradient descent updates satisfy a linear convergence guarantee:*

$$\mathcal{L}(t+1) \leq \left(1 - \frac{\eta \lambda_0}{2}\right) \mathcal{L}(t),$$

*where $\eta = \mathcal{O}\left(\frac{\lambda_0}{n^3 d^2 g^6}\right)$ is the learning rate and $\lambda_0 = \lambda_{\min}(\boldsymbol{H}^\infty)$ is the minimum eigenvalue of the infinite-width kernel.*

**Proof Sketch.** The proof of Theorem 4.2, detailed in Appendix B, proceeds by induction. The core idea is to show that the network operates in the "lazy training" regime where the tangent kernel remains stable. We first expand the loss at step $t + 1$:

$$\|\boldsymbol{y} - \boldsymbol{u}(t+1)\|_2^2 = \|\boldsymbol{y} - \boldsymbol{u}(t)\|_2^2 - 2(\boldsymbol{y} - \boldsymbol{u}(t))^T(\boldsymbol{u}(t+1) - \boldsymbol{u}(t)) + \|\boldsymbol{u}(t+1) - \boldsymbol{u}(t)\|_2^2$$

. The change in the output, $\boldsymbol{u}(t+1) - \boldsymbol{u}(t)$, can be approximated by a first-order Taylor series, which relates it to the tangent kernel at time $t$, $\boldsymbol{H}(t)$ Jacot et al. (2018). Using stability Lemmas below, 4.3, 4.4, and 4.5, we show that $\boldsymbol{H}(t)$ remains close to the deterministic, infinite-width kernel $\boldsymbol{H}^\infty$. This stability allows us to bound the terms in the expansion and demonstrate a consistent linear decrease in the loss at each step.

**Lemma 4.3** (Coefficient Stability). *Under the assumptions of Theorem 4.2, the first-layer coefficients remain in a small neighborhood of their initialization values throughout training. That is, $|\alpha_{ijk}(t) - \alpha_{ijk}(0)| \leq R$, where $R = \mathcal{O}\left(\frac{g\sqrt{n}}{\lambda_0 \sqrt{m}} \|\boldsymbol{u}(0) - \boldsymbol{y}\|_2\right)$.*

**Lemma 4.4** (Kernel Stability over Time). *With high probability, the distance between the tangent kernel at time $t$ and at initialization is bounded: $\|\boldsymbol{H}(t) - \boldsymbol{H}(0)\|_2 \leq 2n^2 d^2 g^4 R$.*

**Lemma 4.5** (Initial Kernel Concentration). *With high probability, the distance between the initial tangent kernel and the infinite-width kernel is bounded: $\|\boldsymbol{H}(0) - \boldsymbol{H}^\infty\|_2 \leq \frac{dg^3 n}{\sqrt{m}} \sqrt{\log\left(\frac{2n^2}{\delta}\right)}$.*

## 4.2 LABEL-DEPENDENT CONVERGENCE RATE

Next, we refine the convergence rate to show its dependency on the structure of the data labels.

**Theorem 4.6** (Label-Dependent Convergence Bound). *Under the same conditions as Theorem 4.2, let the eigendecomposition of the KAN-TK be $\boldsymbol{H}^\infty = \sum_{i=1}^n \lambda_i \boldsymbol{v}_i \boldsymbol{v}_i^T$. Then the error vector at time $t$ can be bounded as:*

$$\|\boldsymbol{y} - \boldsymbol{u}(t)\|_2 \leq \sqrt{\sum_{i=1}^n (1 - \eta\lambda_i)^{2t} (\boldsymbol{v}_i^T \boldsymbol{y})^2} \pm \epsilon$$

*where $\epsilon$ is a small error term that vanishes as $m \to \infty$.*

**Proof Sketch.** To prove Theorem 4.6, we start with the gradient descent update rule and show that the change in the output can be approximated as $\boldsymbol{u}(t+1) - \boldsymbol{u}(t) \approx -\eta\boldsymbol{H}^\infty(\boldsymbol{u}(t) - \boldsymbol{y})$. This allows us to express the error vector at step $t + 1$ as a recurrence relation: $(\boldsymbol{u}(t+1) - \boldsymbol{y}) \approx (\boldsymbol{I} - \eta\boldsymbol{H}^\infty)(\boldsymbol{u}(t) - \boldsymbol{y})$. Unrolling this recurrence yields $\boldsymbol{u}(t) - \boldsymbol{y} \approx -(\boldsymbol{I} - \eta\boldsymbol{H}^\infty)^t(\boldsymbol{u}(0) - \boldsymbol{y})$. By assuming a small initialization variance $\sigma^2$, the initial output $\|\boldsymbol{u}(0)\|_2$ is negligible compared to $\|\boldsymbol{y}\|_2$. Taking the norm and applying the eigendecomposition of $\boldsymbol{H}^\infty$ gives the desired label-dependent bound. The full proof is deferred to Appendix C.

**Remark 1** (Eigenstructure and Convergence Speed). *Theorem 4.6 demonstrates that the components of the error aligned with eigenvectors ($\boldsymbol{v}_i$) corresponding to large eigenvalues ($\lambda_i$) decay the fastest. Consequently, if the label vector $\boldsymbol{y}$ has a strong projection onto these top eigenvectors (i.e., the labels have a structure that the kernel is well-suited to learn), the overall convergence will be much faster than if the labels were random or aligned with eigenvectors of small eigenvalues.*

## 5 EXPERIMENTS

We conduct a series of experiments using a two-layer KAN with RBF basis functions to validate our theoretical claims. Our implementation is based on the FastKAN architecture (Li, 2024). In all experiments, we train only the first-layer coefficients using full-batch gradient descent, keeping the second-layer coefficients fixed after their random initialization. Additional experimental results are provided in Appendix D.

### 5.1 CONVERGENCE RATE VS. NETWORK WIDTH

To validate Theorem 4.2 and the underlying "lazy training" phenomenon, we study how the hidden layer width $m$ influences convergence.

**Setup.** We generate a synthetic dataset with $n = 100$ samples in $d = 100$ dimensions, where each feature is drawn from a standard normal distribution. Labels are drawn independently from $\mathcal{N}(0, 1)$ to create a challenging learning task. We train KANs with varying hidden widths ($m \in \{500, 1000, 2000, 4000, 8000, 16000, 32000\}$) for 5000 epochs. Each configuration is evaluated over three independent runs, and the resulting training errors are reported.

**Results.** Figure 2a reports the training error across epochs. As predicted by Theorem 4.2, larger widths $m$ yield faster convergence. Figure 2b shows the maximum distance of the weight coefficients from initialization, $\|\boldsymbol{\alpha}(t) - \boldsymbol{\alpha}(0)\|_\infty$. As $m$ increases, the weights travel shorter distances, empirically confirming the "lazy training" assumption in Lemma 4.3.

### 5.2 IMPACT OF LABEL STRUCTURE ON CONVERGENCE

We now empirically evaluate Theorem 4.6, which predicts that the convergence rate of gradient descent is determined by how the label vector $\boldsymbol{y}$ aligns with the eigenspectrum of the KAN-TK.

**Setup for Figure 3a.** We generate a one-dimensional dataset with $n = 50$ points sampled uniformly from $[-1, 1]$. After computing the infinite-width KAN-TK $\boldsymbol{H}^\infty$, we project several label

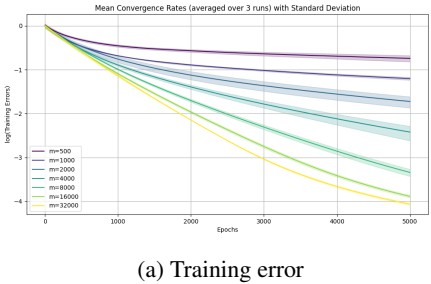
(a) Training error

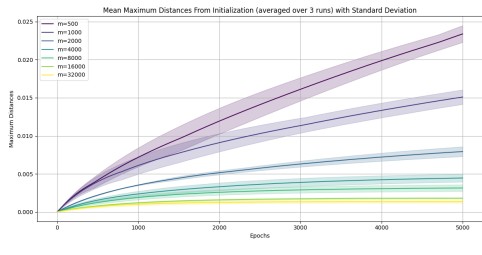
(b) Weight distance from initialization

Figure 2: Convergence behavior across hidden widths $m$. (a) Training error decreases faster for wider networks. (b) Wider networks exhibit smaller deviations from initialization, consistent with the lazy training regime.

configurations onto its eigenvectors. We compare *structured* labels of the form

$$y = \frac{\sin^2(0.7\,x/2)}{\sin^2(x/2)}\,, \tag{1}$$

with *random* labels drawn independently from $\mathcal{N}(0,1)$.

**Setup for Figure 3b.** We conduct a second experiment on a similar one-dimensional dataset with $n = 30$ uniformly spaced points in $[-1, 1]$. We evaluate three label configurations:

1. *Structured*, given by Eq. equation 1;
2. *Random*, sampled i.i.d. from $\mathcal{N}(0,1)$;
3. *Anti-structured*, defined as the eigenvector of $\boldsymbol{H}^\infty$ associated with its smallest eigenvalue.

For all settings, we train a two-layer RBF-based KAN with hidden width $m = 5000$, updating only the first-layer coefficients for 3000 epochs using full-batch gradient descent.

**Results.** Figure 3a illustrates the projections of the structured and random label vectors onto the eigenbasis of $\boldsymbol{H}^\infty$. The structured labels concentrate most of their energy on the top eigenvectors, whereas random labels distribute their mass more uniformly across the spectrum. Figure 3b shows the resulting optimization dynamics: networks trained on structured labels converge the fastest, random labels converge at a moderate rate, and anti-structured labels converge the slowest. Together, these observations provide strong empirical support for the label-dependent convergence behavior predicted by Theorem 4.6.

## 6 COMPARISON AND DISCUSSION

We now compare the complexity of our proposed training scheme with two key benchmarks: (1) standard two-layer neural networks Du et al. (2019), and (2) two-layer KANs where both layers are trained (Gao & Tan, 2025). This analysis highlights the trade-offs between parameter efficiency, stability, and convergence speed.

### 6.1 PARAMETER AND WIDTH COMPARISON

Table 1 summarizes the asymptotic requirements on hidden layer width and the number of trainable parameters needed to guarantee convergence to a global minimum.

Our method substantially reduces the required network width ($m$) compared to standard ReLU-activated Neural Networks (NNs) while employing the same Tangent Kernel (TK) stability analysis methodology used for examining neural networks in the overparameterized regime (Du et al., 2019). Specifically, classical two-layer NNs (often using ReLU) require a width of $m = \mathcal{O}(n^6)$ to guarantee convergence, whereas our first-layer-only Kolmogorov-Arnold Network (KAN) achieves this

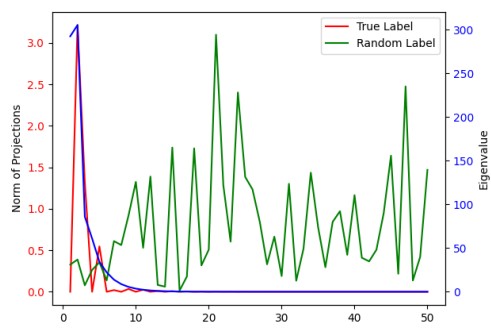 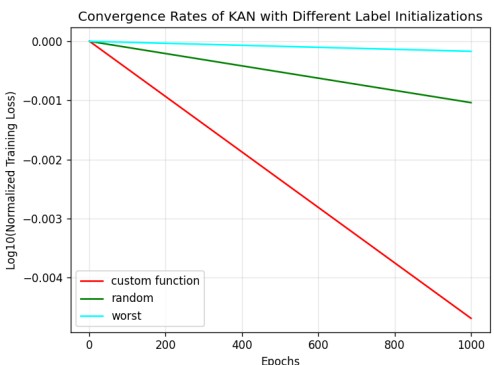

(a) Projection of labels onto the eigenspectrum.    (b) Convergence by label structure.

Figure 3: Effect of label structure on convergence. (a) Structured labels align with top eigenvectors, whereas random labels distribute across the spectrum. (b) Training converges fastest for structured labels, slower for random labels, and slowest for anti-structured labels.

Table 1: Comparison of required hidden layer width and number of trainable parameters for global convergence guarantees.

| Network Type | Hidden Layer Width ($m$) | Trainable Parameters |
|---|---|---|
| Neural Network (Du et al., 2019) | $\mathcal{O}\left(\dfrac{n^6}{\lambda_0^4 \delta^3}\right)$ | $\mathcal{O}\left(\dfrac{n^6 d}{\lambda_0^4 \delta^3}\right)$ |
| KAN (Both Layers) (Gao & Tan, 2025) | $\tilde{\mathcal{O}}\left(\dfrac{g^9 n^3}{\lambda_0^4}\right)$ | $\tilde{\mathcal{O}}\left(\dfrac{g^{10} n^3 d}{\lambda_0^4}\right)$ |
| KAN (First Layer Only) (Ours) | $\mathcal{O}\left(\dfrac{d^2 g^6 n^2}{\lambda_0^2}\right)$ | $\mathcal{O}\left(\dfrac{d^3 g^7 n^2}{\lambda_0^2}\right)$ |

guarantee with $m = \mathcal{O}(n^2)$, highlighting a parameter-efficiency advantage. This enhanced efficiency stems directly from the superior expressive power of the learnable basis functions (such as polynomials) inherent in KAN architectures, which alleviate the need for extremely wide layers.

Compared to training both layers of a KAN, our method achieves improved stability with respect to $\lambda_0$. In particular, the dependence on the minimum eigenvalue of the tangent kernel, $\lambda_0$, improves from $\lambda_0^{-4}$ to $\lambda_0^{-2}$. This weaker dependence is advantageous because $\lambda_0$ can be very small in practice, and guarantees that are less sensitive to its value are therefore more robust. For instance, if $\lambda_0$ decreases by a factor of $k$, a neural network would require $k^4$ times more width to maintain convergence, whereas a KAN would require only $k^2$ times more width. The trade-off is that our bounds introduce a stronger dependence on the input dimension $d$ and the number of basis functions $g$. Nonetheless, since the dataset size $n$ typically dominates in practical settings, we regard this as a favorable trade-off between stability and parameter scaling.

**Remark 2** (Why KANs Achieve Better Width Scaling than MLPs). *KANs require only $O(n^2)$ width for kernel concentration and convergence, whereas two-layer ReLU networks typically need $O(n^6)$. The fundamental reason is the smooth and stable nature of KAN features during training. As emphasized in the original KAN paper, KANs replace neuron-level activations with learnable univariate spline functions along edges. As a result, intermediate representations are compositions of smooth one-dimensional functions rather than brittle, sign-dependent ReLU activations. This smoothness ensures that the Neural Tangent Kernel (NTK) of a KAN depends only on bounded derivatives of these splines and involves at most pairwise interactions between samples, yielding concentration with width scaling that is only quadratic in the dataset size. In contrast, classical ReLU networks must maintain stability of discrete activation patterns during training. NTK analyses (e.g., (Du et al., 2019)) show that preventing activation-pattern flips requires controlling higher-order interactions among samples, which amplifies into the $O(n^6)$ width requirement. Thus, the structural design*

Table 2: Comparison of required learning rates for guaranteed convergence.

| Network Type | Learning Rate ($\eta$) |
|---|---|
| Neural Network (Du et al., 2019) | $\mathcal{O}\left(\frac{\lambda_0}{n^2}\right)$ |
| KAN (Both Layers) (Gao & Tan, 2025) | $\mathcal{O}\left(\frac{1}{g}\right)$ |
| KAN (First Layer Only) (Ours) | $\mathcal{O}\left(\frac{\lambda_0}{n^3 d^2 g^6}\right)$ |

*of KANs—learnable smooth functions on edges, aligned with the Kolmogorov–Arnold representation—eliminates the combinatorial instability inherent to ReLU networks and leads directly to the improved $O(n^2)$ scaling.*

**Remark 3** (More Advanced Methods). *Recent advances by Polaczyk & Cyranka (2023) provide a refined analysis of overparameterized networks using Clarke subdifferentials and differential inclusions, yielding a tighter width bound of $\mathcal{O}(n^{1.25})$ for global convergence. While adapting these non-linear tools to KANs offers a promising path for tighter theoretical guarantees, we maintain our benchmark against Du et al. (2019) to ensure methodological consistency with foundational results in the field.*

## 6.2 CONVERGENCE RATE COMPARISON

While our approach is more parameter-efficient and stable, it requires a smaller learning rate, which in turn leads to slower convergence. The key difference lies in the allowable step size $\eta$.

As shown in Table 2, our method requires a smaller step size than either of the benchmarks. Since the linear convergence rate scales with $\eta\lambda_0$, this smaller $\eta$ results in slower learning. This trade-off is expected: by simplifying the optimization to only the first layer, we obtain stronger guarantees on parameter efficiency and stability, at the expense of convergence speed.

## 7 CONCLUSION

This work provides a theoretical analysis of the optimization dynamics of two-layer Kolmogorov-Arnold Networks in the overparameterized regime. By focusing on a simplified setting where only the first layer is trained, we prove that gradient descent converges to a global minimum, achieving zero training error. We also provide a fine-grained, label-dependent convergence rate that connects the optimization speed to the intrinsic structure of the learning task. Our results demonstrate that KANs are not only more interpretable but also significantly more parameter-efficient than classical neural networks with ReLU activations, requiring a polynomially smaller hidden layer width ($m = O(n^2)$ vs. $m = O(n^6)$) to guarantee convergence.

Our analysis opens several promising avenues for future research. An immediate next step is to extend this theoretical framework to deep KANs to understand the role of depth in the training dynamics and convergence rates. Another important direction is to analyze the behavior of KANs under more practical, stochastic optimization algorithms like Adam. Furthermore, exploring alternative theoretical methodologies beyond the tangent kernel framework, like the one in Polaczyk & Cyranka (2023), is crucial for deriving tighter convergence bounds. We can also examine the interpretability of KANs specifically within the overparameterized regime, connecting theoretical guarantees with explanatory power. Additional research should focus on deriving closed-form expressions for the KAN Tangent Kernel for multi-dimensional inputs and other basis functions, which would provide deeper insights into different KAN architectures. Finally, we must also examine the impact of various initialization techniques on the performance and theoretical guarantees of KANs in the overparameterized setting.

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

## A TANGENT KERNELS

### A.1 BACKGROUND ON TANGENT KERNELS

The tangent kernel is a key concept for analyzing the training dynamics of overparameterized networks. Formally, for a model $f_{\boldsymbol{\theta}}(\boldsymbol{x})$ with parameters $\boldsymbol{\theta}$, the tangent kernel is defined as

$$H_{ij} = \langle \nabla_{\boldsymbol{\theta}} f_{\boldsymbol{\theta}}(\boldsymbol{x}_i), \nabla_{\boldsymbol{\theta}} f_{\boldsymbol{\theta}}(\boldsymbol{x}_j) \rangle$$

where $\boldsymbol{x}_i, \boldsymbol{x}_j$ are data samples. Intuitively, $H$ measures how similarly parameter updates induced by different data points affect the model output.

In the so-called *lazy training regime*, which arises when the network is sufficiently wide, the tangent kernel remains nearly constant throughout training. This stability means that the nonlinear training dynamics of the network can be closely approximated by a linear model whose evolution is governed by this fixed kernel. As a consequence, gradient descent on the network is equivalent to performing kernel regression with the tangent kernel (Jacot et al., 2018).

For standard neural networks, this leads to the well-known Neural Tangent Kernel (NTK). In our case, where we focus on two-layer Kolmogorov–Arnold Networks (KANs) with only the first-layer coefficients trained, the analogous object is the *KAN Tangent Kernel (KAN-TK)*. The KAN-TK captures the interaction between input features and learnable basis-function coefficients. In the infinite-width limit ($m \to \infty$), we can derive a deterministic closed-form expression for KAN-TK when using RBF basis functions, which we employ throughout our experiments.

**Finite- and infinite-width kernels.** If we run an optimization algorithm, then the parameters $\boldsymbol{\theta}$ evolve with time, making the tangent kernel time dependent. We denote the kernel at step $t$ by

$$\boldsymbol{H}(t) \;=\; \big(H_{ij}(t)\big)_{i,j=1}^{n}$$

which is computed from the gradients at that point in training. If the network is initialized randomly, then $\boldsymbol{H}(0)$ is itself a random matrix. Its expectation over random initialization defines the *infinite-width tangent kernel*, denoted by $\boldsymbol{H}^{\infty}$.

**Networks Act Like Kernel Ridge Regression.** To see why wide neural networks effectively behave like kernel methods, note that in the lazy training regime the features $\nabla_{\boldsymbol{\theta}} f_{\boldsymbol{\theta}}(\boldsymbol{x}_i)$ remain nearly constant during training. This means that the model output at time $t$ can be approximated by a linear expansion around initialization:

$$f_{\boldsymbol{\theta}(t)}(\boldsymbol{x}) \;\approx\; f_{\boldsymbol{\theta}(0)}(\boldsymbol{x}) + \nabla_{\boldsymbol{\theta}} f_{\boldsymbol{\theta}(0)}(\boldsymbol{x})^{T} \big(\boldsymbol{\theta}(t) - \boldsymbol{\theta}(0)\big).$$

Since the gradient features are fixed, learning reduces to finding linear coefficients on this (very high-dimensional) feature map. By the representer theorem, this is equivalent to solving a kernel ridge regression problem with kernel matrix $\boldsymbol{H}$, where each entry $H_{ij}$ measures the similarity between features induced by samples $\boldsymbol{x}_i$ and $\boldsymbol{x}_j$.

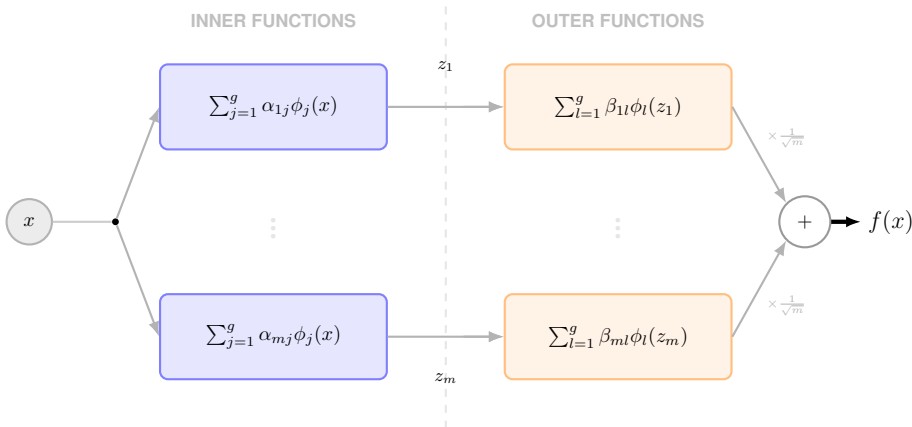

Figure 4: Architecture of a two-layer Kolmogorov-Arnold Network (KAN) with a one dimensional input. The inner functions (blue) map inputs to intermediate latent variables $z$, which are then processed by outer functions (orange) and aggregated.

In other words, training an infinitely wide neural network with gradient descent is mathematically the same as performing kernel regression with its tangent kernel. The nonlinearity of the original network is thus captured entirely through the structure of $\boldsymbol{H}^{\infty}$, while the optimization itself is no more complicated than linear regression in feature space.

**Connection to training dynamics.** One of the main advantages of tangent kernels is that they allow us to describe the network's dynamics explicitly. For example, under gradient flow optimization Du et al. (2019), the output vector evolves according to

$$\frac{d\boldsymbol{u}(t)}{dt} = \boldsymbol{H}(t)\left(\boldsymbol{y} - \boldsymbol{u}(t)\right).$$

This shows that the convergence behavior of the network is governed entirely by the spectral properties of $\boldsymbol{H}(t)$ (or $\boldsymbol{H}^{\infty}$ in the infinite-width case).

Thus, the tangent kernel viewpoint bridges the gap between the nonlinear training of KANs and a tractable kernel regression framework, offering both analytical insights and practical tools for understanding their optimization behavior.

## A.2    PROOF OF PROPOSITION 3.1

Figure 4 illustrates the two-layer Kolmogorov-Arnold Network (KAN) in the special case of a one-dimensional input. This schematic clarifies the roles of the $\alpha_{ij}$ and $\beta_{il}$ coefficients, the intermediate activations $z_i$, and the final scaled aggregation $\frac{1}{\sqrt{m}}\sum_i \sum_l \beta_{il}\phi_l(z_i)$ that produces the network output $f(x)$. The subsequent analysis in this appendix derives the infinite-width kernel $\boldsymbol{H}^{\infty}$ associated with this architecture by decomposing it into the contributions from the $\beta$ parameters ($\boldsymbol{H}_1^{\infty}$) and the $\alpha$ parameters ($\boldsymbol{H}_2^{\infty}$).

By definition, we have:

$$H_{qr}^{\infty} = \overbrace{\langle \frac{\partial f(x^q)}{\partial \beta}, \frac{\partial f(x^r)}{\partial \beta}\rangle}^{(H_1^{\infty})_{qr}} + \overbrace{\langle \frac{\partial f(x^q)}{\partial \alpha}, \frac{\partial f(x^r)}{\partial \alpha}\rangle}^{(H_2^{\infty})_{qr}} \tag{2}$$

First, we compute the $\boldsymbol{H}_1^\infty$ term. From the definition of our network, we know that $\frac{\partial f(x)}{\partial \beta_{ij}} = \frac{1}{\sqrt{m}} \phi_j(z_i)$. From this, we can conclude:

$$(H_1^\infty)_{qr} = \frac{1}{m} \sum_{i=1}^m \sum_{j=1}^g \phi_j(z_i^q) \phi_j(z_i^r)$$

$$= \mathbb{E}\left[ \sum_{j=1}^g \phi_j(z^q) \phi_j(z^r) \right] \tag{3}$$

where the second line follows from the law of large numbers as $m \to \infty$. We can simplify this as:

$$(H_1^\infty)_{qr} = \mathbb{E}\left[ \sum_{j=1}^g \exp\left( -\frac{(z^q - \mu_j)^2 + (z^r - \mu_j)^2}{2\sigma^2} \right) \right]$$

$$= \sum_{j=1}^g \mathbb{E}\left[ \exp\left( -\frac{2\mu_j^2 + (z^q)^2 + (z^r)^2 - 2\mu_j(z^q + z^r)}{2\sigma^2} \right) \right]$$

$$= \exp\left( -\frac{\mu_j^2}{\sigma^2} \right) \sum_{j=1}^g \mathbb{E}\left[ \exp\left( -\frac{\overbrace{(z^q)^2 + (z^r)^2 - 2\mu_j(z^q + z^r)}^{S_j^{qr}}}{2\sigma^2} \right) \right] \tag{4}$$

Since $z^q = \sum_{l=1}^g \alpha_l \phi_l(x^q)$ and $\alpha \sim \mathcal{N}(0, \boldsymbol{I}_g)$, we can write $S_j^{qr} = \alpha^T \boldsymbol{A}^{qr} \alpha + \mu_j(\boldsymbol{b}^{qr})^T \alpha$ where

$$A_{kl}^{qr} = \phi_l(x^q)\phi_k(x^q) + \phi_l(x^r)\phi_k(x^r) \quad \text{and} \quad b_l^{qr} = -2(\phi_l(x^q) + \phi_l(x^r)) \tag{5}$$

Using the moment generating function for a quadratic form of Gaussian random variables Mathai & Provost (1992), we get:

$$\mathbb{E}\left[ \exp\left( t S_j^{qr} \right) \right] = \frac{\exp\left( \frac{t^2 \mu_j^2}{2} (\boldsymbol{b}^{qr})^T (\boldsymbol{I} - 2t\boldsymbol{A}^{qr})^{-1} \boldsymbol{b}^{qr} \right)}{\sqrt{\det\left( \boldsymbol{I} - 2t\boldsymbol{A}^{qr} \right)}} \tag{6}$$

Setting $t = -1/(2\sigma^2)$ gives:

$$(H_1^\infty)_{qr} = \sum_{j=1}^g \frac{\exp\left( -\frac{\mu_j^2}{\sigma^2} \right)}{\sqrt{\det\left( \boldsymbol{I} + \frac{1}{\sigma^2} \boldsymbol{A}^{qr} \right)}} \exp\left( \frac{\mu_j^2}{8\sigma^4} (\boldsymbol{b}^{qr})^T \left( \boldsymbol{I} + \frac{1}{\sigma^2} \boldsymbol{A}^{qr} \right)^{-1} \boldsymbol{b}^{qr} \right) \tag{7}$$

Next, we compute $\boldsymbol{H}_2^\infty$. The derivative with respect to $\alpha_{ij}$ is:

$$\frac{\partial f(x)}{\partial \alpha_{ij}} = \frac{1}{\sqrt{m}} \sum_{l=1}^g \beta_{il} \phi_l'(z_i) \phi_j(x) \tag{8}$$

This leads to:

$$(H_2^\infty)_{qr} = \mathbb{E}\left[ \sum_{s,l,j=1}^g \beta_l \beta_s \phi_l'(z^q) \phi_s'(z^r) \phi_j(x^q) \phi_j(x^r) \right]$$

$$= \sum_{j,l=1}^g \mathbb{E}\left[ \phi_l'(z^q) \phi_l'(z^r) \phi_j(x^q) \phi_j(x^r) \right] \tag{9}$$

where the second line follows because $\mathbb{E}[\beta_l \beta_s] = \delta_{ls}$. Since $\phi_l'(z) = -\frac{z - \mu_l}{\sigma^2} \phi_l(z)$, we have:

$$(H_2^\infty)_{qr} = \sum_{j,l=1}^g \frac{\phi_j(x^q)\phi_j(x^r)}{\sigma^4} \mathbb{E}\left[ (z^q - \mu_l)(z^r - \mu_l)\phi_l(z^q)\phi_l(z^r) \right] \tag{10}$$

The expectation term can be written as:

$$\mathbb{E}\left[(z^q - \mu_l)(z^r - \mu_l)\exp\left(-\frac{(z^q - \mu_l)^2 + (z^r - \mu_l)^2}{2\sigma^2}\right)\right] \tag{11}$$

Let $Z_l^{qr}(t) = \mathbb{E}[\exp(tS_l^{qr})]$. We can relate the expectation to derivatives of $Z_l^{qr}(t)$ with respect to the components of $\boldsymbol{b}^{qr}$.

$$\frac{\partial Z_l^{qr}(t)}{\partial b_s^{qr}} = t\mu_l \mathbb{E}[\alpha_s \exp(tS_l^{qr})] \tag{12}$$

$$\frac{\partial^2 Z_l^{qr}(t)}{\partial b_p^{qr}\partial b_s^{qr}} = (t\mu_l)^2 \mathbb{E}[\alpha_s\alpha_p \exp(tS_l^{qr})] \tag{13}$$

And we know that $(z^q - \mu_l)(z^r - \mu_l) = \sum_{s,p}\alpha_s\alpha_p\phi_s(x^q)\phi_p(x^r) + \mu_l^2 + \mu_l\sum_s b_s^{qr}\alpha_s$. Putting these pieces together, we can express the expectation in terms of $Z_l^{qr}(t)$ and its derivatives:

$$\mathbb{E}\left[(z^q - \mu_l)(z^r - \mu_l)\exp\{tS_l^{qr}\}\right] = \sum_{s,p}\frac{\phi_s(x^q)\phi_p(x^r)}{(t\mu_l)^2}\frac{\partial^2 Z_l^{qr}(t)}{\partial b_p^{qr}\partial b_s^{qr}}$$

$$+ \mu_l^2 Z_l^{qr}(t) + \sum_s \frac{b_s^{qr}}{t}\frac{\partial Z_l^{qr}(t)}{\partial b_s^{qr}} \tag{14}$$

By defining $\boldsymbol{G}^{qr} = (\boldsymbol{I} - 2t\boldsymbol{A}^{qr})^{-1}$ and $T_l^{qr} = \exp\left(\frac{t^2\mu_l^2}{2}(\boldsymbol{b}^{qr})^T(\boldsymbol{I} - 2t\boldsymbol{A}^{qr})^{-1}\boldsymbol{b}^{qr}\right)$, we can find closed forms for the derivatives of $Z_l^{qr}(t)$. Substituting these back gives the final expression for $(H_2^\infty)_{qr}$, which completes the proof. We have the following:

$$\frac{\partial Z_l^{qr}(t)}{\partial b_s^{qr}} = \frac{t^2\mu_l^2((\boldsymbol{I} - 2t\boldsymbol{A}^{qr})^{-1}\boldsymbol{b}^{qr})_s}{\sqrt{\det(\boldsymbol{I} - 2t\boldsymbol{A}^{qr})}}T_l^{qr} \tag{15}$$

$$\frac{\partial^2 Z_l^{qr}(t)}{\partial b_p^{qr}\partial b_s^{qr}} = \frac{t^2\mu_l^2((\boldsymbol{I} - 2t\boldsymbol{A}^{qr})^{-1})_{sp}}{\sqrt{\det(\boldsymbol{I} - 2t\boldsymbol{A}^{qr})}}T_l^{qr}$$

$$+ \frac{t^4\mu_l^4((\boldsymbol{I} - 2t\boldsymbol{A}^{qr})^{-1}\boldsymbol{b}^{qr})_s((\boldsymbol{I} - 2t\boldsymbol{A}^{qr})^{-1}\boldsymbol{b}^{qr})_p}{\det(\boldsymbol{I} - 2t\boldsymbol{A}^{qr})}T_l^{qr} \tag{16}$$

By defining:

$$Y_{sl}^{qr} = \frac{1}{t}\frac{\partial Z_l^{qr}(t)}{\partial b_s^{qr}} = t\mu_l^2\sqrt{\det(\boldsymbol{G}^{qr})}(\boldsymbol{G}^{qr}\boldsymbol{b}^{qr})_s T_l^{qr} \tag{17}$$

$$X_{psl}^{qr} = \frac{1}{t^2\mu_l^2}\frac{\partial^2 Z_l^{qr}(t)}{\partial b_p^{qr}\partial b_s^{qr}} = \sqrt{\det(\boldsymbol{G}^{qr})}(\boldsymbol{G}^{qr})_{sp}T_l^{qr} + t^2\mu_l^2\det(\boldsymbol{G}^{qr})(\boldsymbol{G}^{qr}\boldsymbol{b}^{qr})_s(\boldsymbol{G}^{qr}\boldsymbol{b}^{qr})_p T_l^{qr} \tag{18}$$

we can write:

$$\mathbb{E}\{(z^q - \mu_l)(z^r - \mu_l)\exp\{tS_l^{qr}\}\} = \sum_{s,p}\phi_s(x^q)\phi_p(x^r)X_{psl}^{qr} + \mu_l^2 Z_l^{qr} + \sum_s b_s^{qr}Y_{sl}^{qr} \tag{19}$$

Substituting this back into the expression for $(\boldsymbol{H}_2^\infty)_{qr}$ gives the final result:

$$(H_2^\infty)_{qr} = \sum_{j,l=1}^g \frac{\phi_j(x^q)\phi_j(x^r)\exp\left(-\frac{\mu_l^2}{\sigma^2}\right)}{\sigma^4}\left\{\sum_{s,p}\phi_s(x^q)\phi_p(x^r)X_{psl}^{qr} + \mu_l^2 Z_l^{qr} + \sum_s b_s^{qr}Y_{sl}^{qr}\right\}$$

$$\tag{20}$$

$\square$

# B PROOF OF THEOREM 4.2

We begin by recalling the two-layer Kolmogorov-Arnold Network (KAN) architecture analyzed in this appendix (see also Figure 1):

$$
\begin{cases}
f(\boldsymbol{x}) = \dfrac{1}{\sqrt{m}} \sum_{p=1}^{m} \sum_{l=1}^{g} \beta_{pl} \phi_l(z_p), \\[2mm]
z_p = \sum_{k=1}^{d} \sum_{j=1}^{g} \alpha_{pjk} \phi_j(x_k).
\end{cases}
\tag{21}
$$

This formulation makes explicit the dependence of the network output $f(\boldsymbol{x})$ on the coefficients $\alpha_{pjk}$ and $\beta_{pl}$, which will be central in the stability analysis that follows.

## B.1 PROOF OF LEMMA 4.3 (COEFFICIENT STABILITY)

By the induction hypothesis we have

$$
\mathcal{L}(t) \leq \left(1 - \frac{\eta \lambda_0}{2}\right) \mathcal{L}(t-1).
$$

Hence

$$
\|\boldsymbol{u}(t) - \boldsymbol{y}\|_2^2 \leq \left(1 - \frac{\eta \lambda_0}{2}\right) \|\boldsymbol{u}(t-1) - \boldsymbol{y}\|_2^2,
$$

which implies

$$
\begin{aligned}
\|\boldsymbol{u}(t) - \boldsymbol{y}\|_2 &\leq \sqrt{1 - \frac{\eta \lambda_0}{2}} \, \|\boldsymbol{u}(t-1) - \boldsymbol{y}\|_2 \\
&\leq \left(1 - \frac{\eta \lambda_0}{4}\right) \|\boldsymbol{u}(t-1) - \boldsymbol{y}\|_2 \qquad \left(\text{since } \sqrt{1-x} \leq 1 - \frac{x}{2} \text{ for } 0 \leq x \leq 1\right) \\
&\leq \left(1 - \frac{\eta \lambda_0}{4}\right)^t \|\boldsymbol{u}(0) - \boldsymbol{y}\|_2.
\end{aligned}
\tag{22}
$$

Now consider the gradient descent update for a single coefficient $\alpha_{ijk}$:

$$
\begin{aligned}
\alpha_{ijk}(t) - \alpha_{ijk}(t-1) &= -\eta \frac{\partial \mathcal{L}(t-1)}{\partial \alpha_{ijk}} \\
&= -\frac{\eta}{\sqrt{m}} \sum_{q=1}^{n} (u_q(t-1) - y_q) \frac{\partial}{\partial \alpha_{ijk}} \left(\sum_{p=1}^{m} \sum_{l=1}^{g} \beta_{pl} \, \phi_l(z_p^q)\right).
\end{aligned}
\tag{23}
$$

Taking absolute values and using $|\phi_l'(\cdot)| \leq 1$ from the assumptions,

$$
\begin{aligned}
|\alpha_{ijk}(t) - \alpha_{ijk}(t-1)| &\leq \frac{\eta}{\sqrt{m}} \sum_{q,p,l} |\phi_l'(z_p^q)| \left|\frac{\partial z_p^q}{\alpha_{ijk}}\right| |u_q(t-1) - y_q| \\
&\leq \frac{\eta}{\sqrt{m}} \sum_{q,p,l} |\phi_l'(z_p^q)| \, |\phi_j(x_k^q) \, \delta_{ip}| \, |u_q(t-1) - y_q| \\
&\leq \frac{\eta g}{\sqrt{m}} \sum_{q=1}^{n} |u_q(t-1) - y_q| \\
&\leq \frac{\eta g \sqrt{n}}{\sqrt{m}} \|\boldsymbol{u}(t-1) - \boldsymbol{y}\|_2,
\end{aligned}
\tag{24}
\tag{25}
$$

where in equation 24 we used $\delta_{ip} = \mathbb{I}\{i = p\}$, and in equation 25 the inequality $\|\boldsymbol{x}\|_1 \leq \sqrt{n} \|\boldsymbol{x}\|_2$ for $\boldsymbol{x} \in \mathbb{R}^n$.

Summing these updates over $\tau = 0$ to $t - 1$,

$$
\begin{aligned}
|\alpha_{ijk}(t) - \alpha_{ijk}(0)| &\le \sum_{\tau=0}^{t-1} |\alpha_{ijk}(\tau+1) - \alpha_{ijk}(\tau)| \\
&\le \eta g \sqrt{\frac{n}{m}} \sum_{\tau=0}^{t-1} \|\boldsymbol{u}(\tau) - \boldsymbol{y}\|_2 \\
&\le \eta g \sqrt{\frac{n}{m}} \sum_{\tau=0}^{t-1} \left(1 - \frac{\eta\lambda_0}{4}\right)^{\tau} \|\boldsymbol{u}(0) - \boldsymbol{y}\|_2 \qquad \text{(by equation 22)} \\
&= \eta g \sqrt{\frac{n}{m}} \|\boldsymbol{u}(0) - \boldsymbol{y}\|_2 \cdot \frac{1 - (1 - \frac{\eta\lambda_0}{4})^t}{1 - (1 - \frac{\eta\lambda_0}{4})} \\
&\le \frac{4g\sqrt{n}}{\lambda_0\sqrt{m}} \|\boldsymbol{u}(0) - \boldsymbol{y}\|_2.
\end{aligned}
\tag{26}
$$

Defining

$$
R := \frac{4g\sqrt{n}}{\lambda_0\sqrt{m}} \|\boldsymbol{u}(0) - \boldsymbol{y}\|_2,
$$

we conclude that $|\alpha_{ijk}(t) - \alpha_{ijk}(0)| \le R$ for all $t$, completing the proof. $\qquad\square$

### B.2 PROOF OF LEMMA 4.4 (KERNEL STABILITY OVER TIME)

By definition, the $(q, r)$ entry of the tangent kernel at time $t$ is

$$
H_{qr}(t) = \left\langle \frac{\partial u_q(t)}{\partial \alpha}, \frac{\partial u_r(t)}{\partial \alpha} \right\rangle.
\tag{27}
$$

From Section B.1 we have already computed

$$
\frac{\partial u_q(t)}{\partial \alpha_{ijk}} = \frac{1}{\sqrt{m}} \sum_{l=1}^{g} \beta_{il}\, \phi_l'(z_i^q(t))\, \phi_j(x_k^q).
\tag{28}
$$

Substituting equation 28 into equation 27 gives

$$
\begin{aligned}
H_{qr}(t) &= \sum_{i=1}^{m} \sum_{j=1}^{g} \sum_{k=1}^{d} \frac{1}{m} \left( \sum_{l=1}^{g} \beta_{il}\, \phi_l'(z_i^q(t))\, \phi_j(x_k^q) \right) \left( \sum_{s=1}^{g} \beta_{is}\, \phi_s'(z_i^r(t))\, \phi_j(x_k^r) \right) \\
&= \frac{1}{m} \sum_{i,j,k,l,s} \beta_{il}\, \beta_{is}\, \phi_l'(z_i^q(t))\, \phi_j(x_k^q)\, \phi_s'(z_i^r(t))\, \phi_j(x_k^r).
\end{aligned}
\tag{29}
$$

Therefore,

$$
\begin{aligned}
|H_{qr}(t) - H_{qr}(0)| &\le \frac{1}{m} \sum_{i,j,k,l,s} |\phi_j(x_k^q)\phi_j(x_k^r)| \left| \phi_l'(z_i^q(t))\phi_s'(z_i^r(t)) - \phi_l'(z_i^q(0))\phi_s'(z_i^r(0)) \right| \\
&\le \frac{1}{m} \sum_{i,j,k,l,s} \left| \phi_l'(z_i^q(t))\phi_s'(z_i^r(t)) - \phi_l'(z_i^q(0))\phi_s'(z_i^r(0)) \right| \\
&\le \frac{1}{m} \sum_{i,j,k,l,s} \left( |\phi_l'(z_i^q(t)) - \phi_l'(z_i^q(0))| + |\phi_s'(z_i^r(t)) - \phi_s'(z_i^r(0))| \right),
\end{aligned}
\tag{30}
$$

where we used $|\phi_j(\cdot)| \le 1$ and $|\phi_l'(\cdot)| \le 1$ and we now for $a, b, c, d \le 1$ we have $|ab - cd| \le |a - c| + |b - d|$.

From the network definition equation 21,

$$|z_i^q(t) - z_i^q(0)| \leq \sum_{k=1}^{d} \sum_{j=1}^{g} |\phi_j(x_k^q)| \, |\alpha_{ijk}(t) - \alpha_{ijk}(0)| \leq gdR, \tag{31}$$

where the last inequality follows from Lemma 4.3 and the bound $|\phi_j(x_k^q)| \leq 1$.

By Assumptions, the second derivative of $\phi_l$ is bounded, hence

$$|\phi_l'(z_i^q(t)) - \phi_l'(z_i^q(0))| \leq |z_i^q(t) - z_i^q(0)| \leq gdR. \tag{32}$$

Substituting equation 32 into equation 30, we obtain

$$|H_{qr}(t) - H_{qr}(0)| \leq \frac{1}{m} \sum_{i,j,k,l,s} 2gdR = 2d^2 g^4 R. \tag{33}$$

Finally, taking matrix norms gives

$$\|\boldsymbol{H}(t) - \boldsymbol{H}(0)\|_2 \;\leq\; \|\boldsymbol{H}(t) - \boldsymbol{H}(0)\|_F \;\leq\; \sum_{q,r=1}^{n} |H_{qr}(t) - H_{qr}(0)| \;\leq\; 2n^2 d^2 g^4 R. \tag{34}$$

This completes the proof. $\qquad\square$

### B.3 PROOF OF LEMMA 4.5 (INITIAL KERNEL CONCENTRATION)

We begin by observing that

$$H_{qr}(0) = \frac{1}{m} \sum_{i,j,k,l,s} \beta_{il}\beta_{is}\phi_l'(z_i^q(0)) \, \phi_j(x_k^q)\phi_s'(z_i^r(0)) \, \phi_j(x_k^r). \tag{35}$$

Since the coefficients $\alpha_{ijk}$ are independent across different $i$, the expression above can be written as the average of $m$ i.i.d. random variables

$$X_i^{qr} = \sum_{j,k,l,s} \beta_{il}\beta_{is}\phi_l'(z_i^q(0)) \, \phi_j(x_k^q)\phi_s'(z_i^r(0)) \, \phi_j(x_k^r). \tag{36}$$

By our assumptions, each variable is bounded in absolute value by

$$|X_i^{qr}| \leq dg^3.$$

Applying Hoeffding's inequality, we obtain

$$\mathbb{P}\big[|H_{qr}(0) - H_{qr}^\infty| \geq \epsilon\big] \leq 2\exp\left(-\frac{m\epsilon^2}{2d^2 g^6}\right). \tag{37}$$

Taking a union bound over all $n^2$ entries of the kernel matrix, it follows that

$$\mathbb{P}\big[\forall q,r \in [n] : |H_{qr}(0) - H_{qr}^\infty| \leq \epsilon\big] \geq 1 - 2n^2 \exp\left(-\frac{m\epsilon^2}{2d^2 g^6}\right). \tag{38}$$

Equivalently, setting

$$\epsilon = \frac{dg^3}{\sqrt{m}}\sqrt{\log\left(\frac{2n^2}{\delta}\right)},$$

we obtain that with probability at least $1 - \delta$,

$$\|\boldsymbol{H}(0) - \boldsymbol{H}^\infty\|_2^2 \leq \|\boldsymbol{H}(0) - \boldsymbol{H}^\infty\|_F^2 \leq \frac{d^2 g^6 n^2}{m}\log\left(\frac{2n^2}{\delta}\right), \tag{39}$$

which establishes Lemma 4.5. $\qquad\square$

### B.4 Proof of Theorem 4.2

**Step 1: Bounding the initial error.** We begin by establishing an upper bound on the initial error. By the triangle inequality,

$$\|\boldsymbol{y} - \boldsymbol{u}(0)\|_2^2 \le 2\|\boldsymbol{y}\|_2^2 + 2\|\boldsymbol{u}(0)\|_2^2. \tag{40}$$

From our assumptions, we have $\|\boldsymbol{y}\|_2^2 \le n$. We now derive a bound for the second term.

**Step 2: Distribution of the initialization.** By the definition of $z_p$ in equation 21, we know that

$$z_p(0) \sim \mathcal{N}\left(0, \sigma^2 \sum_{j,k} \phi_j^2(x_k)\right).$$

Therefore,

$$\mathbb{E}|z_p(0)| = \sqrt{\frac{2}{\pi}}\, \sigma \sqrt{\sum_{j,k} \phi_j^2(x_k)} \lesssim \sigma\sqrt{gd}.$$

Using assumptions, we obtain

$$\mathbb{E}|u_q(0)| \le \frac{1}{\sqrt{m}} \sum_{p,l} \mathbb{E}|\phi_l(z_p(0))|$$

$$\lesssim \sqrt{md}\, g^{3/2}\sigma.$$

By Markov's inequality, with probability at least $1 - \delta$, we have

$$\|\boldsymbol{u}(0)\|_2 \le \|\boldsymbol{u}(0)\|_1 \lesssim \frac{\sqrt{md}}{\delta}\, n g^{3/2}\sigma. \tag{41}$$

Substituting equation 41 into equation 40, we obtain

$$\|\boldsymbol{y} - \boldsymbol{u}(0)\|_2^2 \lesssim n + \frac{md}{\delta^2} n^2 g^3 \sigma^2. \tag{42}$$

**Step 3: Error recursion.** We now analyze the error at step $t + 1$. Expanding the loss, we have

$$\begin{aligned}
\|\boldsymbol{y} - \boldsymbol{u}(t+1)\|_2^2 &= \|\boldsymbol{y} - \boldsymbol{u}(t) - (\boldsymbol{u}(t+1) - \boldsymbol{u}(t))\|_2^2 \\
&= \|\boldsymbol{y} - \boldsymbol{u}(t)\|_2^2 - 2(\boldsymbol{y} - \boldsymbol{u}(t))^\top (\boldsymbol{u}(t+1) - \boldsymbol{u}(t)) + \|\boldsymbol{u}(t+1) - \boldsymbol{u}(t)\|_2^2.
\end{aligned} \tag{43}$$

**Step 4: Defining the error term.** To control the update, define the error term

$$\epsilon_q(t) = u_q(t+1) - u_q(t) + \eta \sum_{r=1}^n H_{qr}(t)\big(u_r(t) - y_r\big). \tag{44}$$

By expanding $H_{qr}(t)$, we obtain

$$\begin{aligned}
\eta \sum_{r=1}^n H_{qr}(t)(u_r(t) - y_r) &= \sum_{p,j,k,l,s,r} \frac{\eta}{m}(u_r(t) - y_r)\beta_{pl}\beta_{ps}\phi_l'(z_p^q(t))\phi_j(x_k^q)\phi_s'(z_p^r(t))\phi_j(x_k^r) \\
&= \frac{-1}{\sqrt{m}} \sum_{p,j,k,l} \beta_{pl}(\alpha_{p,j,k}(t+1) - \alpha_{p,j,k}(t))\phi_l'(z_p^q(t))\phi_j(x_k^q) \\
&= \frac{-1}{\sqrt{m}} \sum_{p,l} \beta_{pl}(z_p^q(t+1) - z_p^q(t))\phi_l'(z_p^q(t)).
\end{aligned} \tag{45}$$

Substituting equation 45 into equation 44, we find

$$\epsilon_q(t) = \frac{1}{\sqrt{m}} \sum_{p,l} \beta_{pl}\Big[\phi_l(z_p^q(t+1)) - \phi_l(z_p^q(t)) - \phi_l'(z_p^q(t))(z_p^q(t+1) - z_p^q(t))\Big].$$

**Step 5: Bounding the Taylor remainder.** By Taylor's theorem and assumptions, we obtain

$$\phi_l(z_p^q(t+1)) - \phi_l(z_p^q(t)) - \phi_l'(z_p^q(t))(z_p^q(t+1) - z_p^q(t)) \leq \frac{1}{2}\left(z_p^q(t+1) - z_p^q(t)\right)^2.$$

Moreover, from equation 25, one can bound

$$|z_p^q(t+1) - z_p^q(t)| \leq \sum_{j,k} |\alpha_{pjk}(t+1) - \alpha_{pjk}(t)||\phi_j(x_k^q)|$$

$$\leq \frac{d\eta\sqrt{n}}{\sqrt{m}}g^2\|\boldsymbol{u}(t) - \boldsymbol{y}\|_2. \tag{46}$$

**Step 6: Bounding the size of the update.** Using equation 46, we find

$$|u_q(t+1) - u_q(t)| \leq \frac{1}{\sqrt{m}}\sum_{p,l}|\phi_l(z_p^q(t+1)) - \phi_l(z_p^q(t))|$$

$$\leq \frac{1}{\sqrt{m}}\sum_{p,l}|z_p^q(t+1) - z_p^q(t)|$$

$$\leq \eta dg^3\sqrt{n}\|\boldsymbol{u}(t) - \boldsymbol{y}\|_2 \tag{47}$$

Thus,

$$\|\boldsymbol{u}(t+1) - \boldsymbol{u}(t)\|_2 \leq \|\boldsymbol{u}(t+1) - \boldsymbol{u}(t)\|_1 \leq \eta dg^3 n^{3/2}\|\boldsymbol{u}(t) - \boldsymbol{y}\|_2$$

Substituting into equation 43, we obtain

$$\|\boldsymbol{y} - \boldsymbol{u}(t+1)\|_2^2 = \|\boldsymbol{y} - \boldsymbol{u}(t)\|_2^2 - 2(\boldsymbol{y} - \boldsymbol{u}(t))^T(-\eta\boldsymbol{H}(\boldsymbol{u}(t) - \boldsymbol{y}) + \boldsymbol{\epsilon}(t)) + \|\boldsymbol{u}(t+1) - \boldsymbol{u}(t)\|_2$$

$$\leq \left(1 - 2\eta\lambda_{\min}(\boldsymbol{H}(t)) + 2\|\boldsymbol{y} - \boldsymbol{u}(t)\|_2\|\boldsymbol{\epsilon}(t)\|_2 + \eta^2 d^2 g^6 n^3\right)\|\boldsymbol{y} - \boldsymbol{u}(t)\|_2^2. \tag{48}$$

**Step 7: Lower bounding the minimum eigenvalue.** We now lower bound $\lambda_{\min}(\boldsymbol{H}(t))$. By Weyl's perturbation inequality Bhatia (2013) and Lemma 4.5, if

$$m = \mathcal{O}\left(\frac{d^2 g^6 n^2}{\lambda_0^2}\log\left(\frac{n}{\delta}\right)\right),$$

then $\|\boldsymbol{H}(0) - \boldsymbol{H}^\infty\|_2 \leq \lambda_0/4$, and hence $\lambda_{\min}(\boldsymbol{H}(0)) \geq \frac{3}{4}\lambda_0$. Furthermore, by Lemma 4.4, if $R = \mathcal{O}(\lambda_0/(n^2 d^2 g^4))$, then

$$\|\boldsymbol{H}(0) - \boldsymbol{H}(t)\|_2 \leq \lambda_0/4.$$

Together these imply

$$\lambda_{\min}(\boldsymbol{H}(t)) \geq \lambda_0/2.$$

**Step 8: Final convergence bound.** Substituting this into equation 48, and using the induction hypothesis $\|\boldsymbol{y} - \boldsymbol{u}(t)\|_2 \leq \|\boldsymbol{y} - \boldsymbol{u}(0)\|_2$, we obtain

$$\|\boldsymbol{y} - \boldsymbol{u}(t+1)\|_2^2 \leq \left(1 - \eta\lambda_0 + c_0\|\boldsymbol{y} - \boldsymbol{u}(0)\|_2\frac{n\eta^2 d^2 g^5}{\sqrt{m}} + \eta^2 d^2 g^6 n^3\right)\|\boldsymbol{y} - \boldsymbol{u}(t)\|_2^2$$

$$\leq \left(1 - \eta\lambda_0 + c_1 n\eta^2 d^2 g^5\sqrt{\frac{n}{m} + \frac{d}{\delta^2}n^2 g^3\sigma^2} + \eta^2 d^2 g^6 n^3\right)\|\boldsymbol{y} - \boldsymbol{u}(t)\|_2^2. \tag{49}$$

Finally, suppose that $\sigma = \mathcal{O}\left(\delta/\sqrt{mng^3 d}\right)$ and $m = \mathcal{O}(n)$. If we choose the learning rate

$$\eta \lesssim \frac{\lambda_0}{n^3 d^2 g^6},$$

then it follows that

$$\|\boldsymbol{y} - \boldsymbol{u}(t+1)\|_2^2 \leq \left(1 - \frac{\eta\lambda_0}{2}\right)\|\boldsymbol{y} - \boldsymbol{u}(t)\|_2^2,$$

which establishes the desired linear convergence rate. $\qquad\square$

## C  PROOF OF THEOREM 4.6

We prove Theorem 4.6 by relying on the lemmas established in Appendix B. Starting from equation 44 we have, for each coordinate,

$$u_q(t+1) - u_q(t) = -\eta \sum_{r=1}^{n} H_{qr}(t)\big(u_r(t) - y_r\big) + \epsilon_q(t). \tag{50}$$

In vector form this yields

$$\boldsymbol{u}(t+1) - \boldsymbol{u}(t) = -\eta \boldsymbol{H}(t)\big(\boldsymbol{u}(t) - \boldsymbol{y}\big) + \boldsymbol{\epsilon}(t), \tag{51}$$

where $\boldsymbol{\epsilon}(t) = [\epsilon_q(t)]_{q=1}^n$ is the coordinate-wise Taylor remainder.

Using Lemmas 4.5 and 4.4 we decompose $\boldsymbol{H}(t) = \boldsymbol{H}^\infty + (\boldsymbol{H}(t) - \boldsymbol{H}^\infty)$ and rewrite equation 51 as

$$\boldsymbol{u}(t+1) - \boldsymbol{u}(t) = -\eta \boldsymbol{H}^\infty\big(\boldsymbol{u}(t) - \boldsymbol{y}\big) - \eta\big(\boldsymbol{H}(t) - \boldsymbol{H}^\infty\big)\big(\boldsymbol{u}(t) - \boldsymbol{y}\big) + \boldsymbol{\epsilon}(t). \tag{52}$$

Define

$$\boldsymbol{\chi}(t) := -\eta\big(\boldsymbol{H}(t) - \boldsymbol{H}^\infty\big)\big(\boldsymbol{u}(t) - \boldsymbol{y}\big).$$

By the triangle inequality and the lemmas controlling $\boldsymbol{H}(0) - \boldsymbol{H}^\infty$ and $\boldsymbol{H}(t) - \boldsymbol{H}(0)$ we obtain the high-probability bound

$$\begin{aligned}
\|\boldsymbol{\chi}(t)\|_2 &\le \eta \|\boldsymbol{H}^\infty - \boldsymbol{H}(t)\|_2 \|\boldsymbol{u}(t) - \boldsymbol{y}\|_2 \\
&\le \eta\Big(\|\boldsymbol{H}^\infty - \boldsymbol{H}(0)\|_2 + \|\boldsymbol{H}(0) - \boldsymbol{H}(t)\|_2\Big)\|\boldsymbol{u}(t) - \boldsymbol{y}\|_2 \\
&\le 2n^2 d^2 g^4 R \|\boldsymbol{u}(t) - \boldsymbol{y}\|_2,
\end{aligned} \tag{53}$$

where the last inequality uses the concrete bounds from Lemmas 4.5 and 4.4 (see main text for the precise dependence on $m$ and $R$).

Set $\boldsymbol{\zeta}(t) := \boldsymbol{\chi}(t) + \boldsymbol{\epsilon}(t)$. Then the one-step recursion becomes

$$\boldsymbol{u}(t+1) - \boldsymbol{y} = (\boldsymbol{I} - \eta \boldsymbol{H}^\infty)\big(\boldsymbol{u}(t) - \boldsymbol{y}\big) + \boldsymbol{\zeta}(t). \tag{54}$$

Unrolling this recursion for $t$ steps gives

$$\begin{aligned}
\boldsymbol{u}(t) - \boldsymbol{y} &= (\boldsymbol{I} - \eta \boldsymbol{H}^\infty)^t\big(\boldsymbol{u}(0) - \boldsymbol{y}\big) + \sum_{\tau=0}^{t-1}(\boldsymbol{I} - \eta \boldsymbol{H}^\infty)^\tau \boldsymbol{\zeta}(t - 1 - \tau) \\
&= -(\boldsymbol{I} - \eta \boldsymbol{H}^\infty)^t \boldsymbol{y} + (\boldsymbol{I} - \eta \boldsymbol{H}^\infty)^t \boldsymbol{u}(0) + \sum_{\tau=0}^{t-1}(\boldsymbol{I} - \eta \boldsymbol{H}^\infty)^\tau \boldsymbol{\zeta}(t - 1 - \tau).
\end{aligned} \tag{55}$$

The first term $-(\boldsymbol{I} - \eta \boldsymbol{H}^\infty)^t \boldsymbol{y}$ is the label-dependent main term in the theorem; we must show the remaining two terms are negligible.

**Bounding the initialization term.**  From equation 41 we have (with high probability)

$$\|(\boldsymbol{I} - \eta \boldsymbol{H}^\infty)^t \boldsymbol{u}(0)\|_2 \le (1 - \eta\lambda_0)^t \|\boldsymbol{u}(0)\|_2 \lesssim (1 - \eta\lambda_0)^t \frac{\sqrt{md}}{\delta} n g^{3/2} \sigma. \tag{56}$$

Thus for small initialization variance $\sigma^2$ the initialization term decays exponentially and is negligible.

**Bounding the accumulated error term.**  Using equation 53 and the bound on $\boldsymbol{\epsilon}(t)$ from Appendix B, we obtain for the accumulated error

$$\begin{aligned}
\Big\|\sum_{\tau=0}^{t-1}(\boldsymbol{I} - \eta \boldsymbol{H}^\infty)^\tau \boldsymbol{\zeta}(t - 1 - \tau)\Big\|_2 &\le \sum_{\tau=0}^{t-1}\|(\boldsymbol{I} - \eta \boldsymbol{H}^\infty)^\tau\|_2 \|\boldsymbol{\zeta}(t - 1 - \tau)\|_2 \\
&\le \sum_{\tau=0}^{t-1}(1 - \eta\lambda_0)^\tau C n^2 d^2 g^4 R \|\boldsymbol{u}(t) - \boldsymbol{y}\|_2 \\
&\lesssim n^2 d^2 g^4 R \sum_{\tau=0}^{t-1}(1 - \eta\lambda_0)^\tau \|\boldsymbol{u}(0) - \boldsymbol{y}\|_2,
\end{aligned} \tag{57}$$

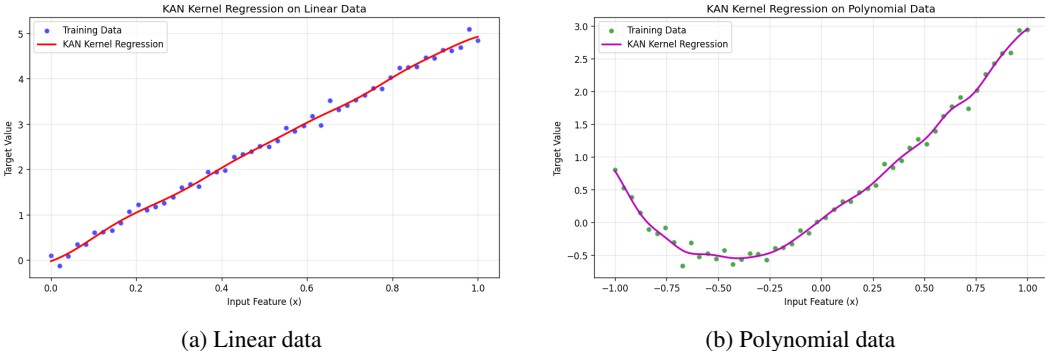

(a) Linear data                      (b) Polynomial data

Figure 5: KAN-TK regression results on synthetic datasets. Figure (a) shows performance on linear data, while Figure (b) shows polynomial data.

where $C$ is an absolute constant absorbed into $\lesssim$ and we used the induction bound $\|\boldsymbol{u}(t) - \boldsymbol{y}\|_2 \le \|\boldsymbol{u}(0) - \boldsymbol{y}\|_2$ in the last line. Substituting the bound equation 42 for $\|\boldsymbol{u}(0) - \boldsymbol{y}\|_2$ yields

$$\Big\| \sum_{\tau=0}^{t-1} (\boldsymbol{I} - \eta \boldsymbol{H}^\infty)^\tau \boldsymbol{\zeta}(t-1-\tau) \Big\|_2 \lesssim n^2 d^2 g^4 R \sqrt{n + \frac{md}{\delta^2} n^2 g^3 \sigma^2} \sum_{\tau=0}^{\infty} (1 - \eta\lambda_0)^\tau$$
$$\lesssim \frac{n^2 d^2 g^4 R}{\eta\lambda_0} \sqrt{n + \frac{md}{\delta^2} n^2 g^3 \sigma^2} \,. \tag{58}$$

**Parameter choices and conclusion.** If we choose the initialization variance and stability radius as

$$\sigma = \mathcal{O}\left( \frac{\delta}{m\sqrt{n g^3 d}} \right), \qquad R = \mathcal{O}\left( \frac{\eta\lambda_0}{n^{5/2} d^2 g^4} \right),$$

then both the initialization term equation 56 and the accumulated error above can be made arbitrarily small. Under these choices the dominant term in equation 55 is $-(\boldsymbol{I} - \eta \boldsymbol{H}^\infty)^t \boldsymbol{y}$, and the label-dependent bound of Theorem 4.6 follows.                                    □

## D  ADDITIONAL EXPERIMENTS

### D.1  ADDITIONAL REGRESSION EXPERIMENTS WITH KAN-TK

As shown in Figure 5, kernel regression with our derived KAN-TK effectively fits both a simple linear function (Figure 5a) and a more complex polynomial function (Figure 5b). This demonstrates that the induced kernel captures the expressive function space of the underlying KAN. To avoid overfitting, we apply Kernel Ridge Regression with a regularization parameter of $\lambda = 0.1$.

**Setup.** For both experiments, we construct datasets of $n = 50$ samples with inputs drawn uniformly from the interval $[-1, 1]$. The linear dataset is generated from

$$y = 5x + \epsilon, \quad \epsilon \sim \mathcal{N}(0, 0.01),$$

while the polynomial dataset is generated from

$$y = 0.5x^4 - 8.6x^3 + 1.32x^2 + 2x + \epsilon, \quad \epsilon \sim \mathcal{N}(0, 0.01).$$

**Results.** Figure 5a shows that the KAN-TK regressor recovers the linear function almost perfectly despite the additive noise. Figure 5b further illustrates that the kernel can fit a substantially more complex nonlinear target function with high accuracy. These results highlight the flexibility of KAN-TK: even with a modest number of samples, it adapts effectively to functions of varying complexity while maintaining robustness through regularization.

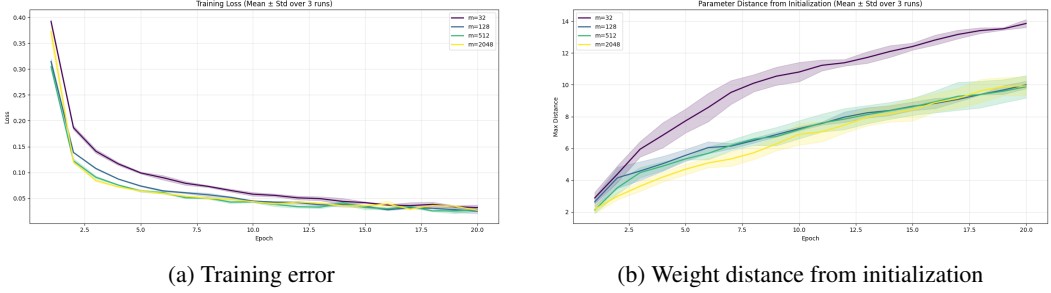

(a) Training error          (b) Weight distance from initialization

Figure 6: Convergence analysis on the MNIST dataset. (a) Training error and (b) $\ell_\infty$ distance of weights from initialization.

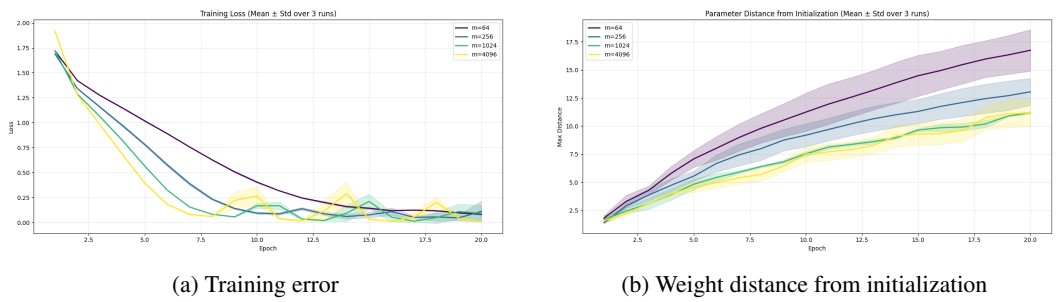

(a) Training error          (b) Weight distance from initialization

Figure 7: Convergence analysis on the CIFAR-10 dataset. (a) Training error and (b) $\ell_\infty$ distance of weights from initialization.

### D.2 EXAMINING RESULTS ON MORE COMPLEX DATASETS

In this section, we evaluate our convergence and distance-from-initialization results on two standard image classification benchmarks: MNIST LeCun et al. (1998) and CIFAR-10 Krizhevsky (2009).

**Setup.** For MNIST, we considered model widths $m \in \{32, 128, 512, 2048\}$, and for CIFAR-10 we used $m \in \{256, 512, 1024, 2048\}$. All experiments were run for 20 epochs using the cross-entropy loss. For each setting, we performed three independent runs to account for randomness and reported the resulting errors.

**Results.** Figure 6 presents the training error and parameter deviation on MNIST, while Figure 7 shows the corresponding results for CIFAR-10. In both datasets, we observe the same empirical trends identified earlier: increasing the network width leads to faster convergence and smaller deviations from the initialization, demonstrating that these behaviors persist even on substantially more complex real-world tasks.

These results, utilizing the cross-entropy loss, also suggest a future direction: examining the behavior of network parameters under cross-entropy to show that they remain close to initialization, consistent with the observations made using the MSE loss in this paper.

### D.3 ADDITIONAL PROJECTIONS OF LABEL VECTORS

In Section 4.2, we analyzed how the structure of the label vector $\boldsymbol{y}$ influences optimization by examining its projection onto the eigenspectrum of the KAN-TK. Here, we extend this analysis to several additional structured label functions to further illustrate the relationship between label-kernel alignment and convergence behavior.

**Setup.** We construct one-dimensional datasets with $n = 50$ samples drawn uniformly from $[-1, 1]$. We consider four structured label functions:

$$y = \exp(x), \qquad y = \ln|x| + x^2 + 1, \qquad y = x\left(1 - \tfrac{x}{3}\right)^{-1}, \qquad y = \sin^{-1}\left(0.4\sin(x)\right).$$

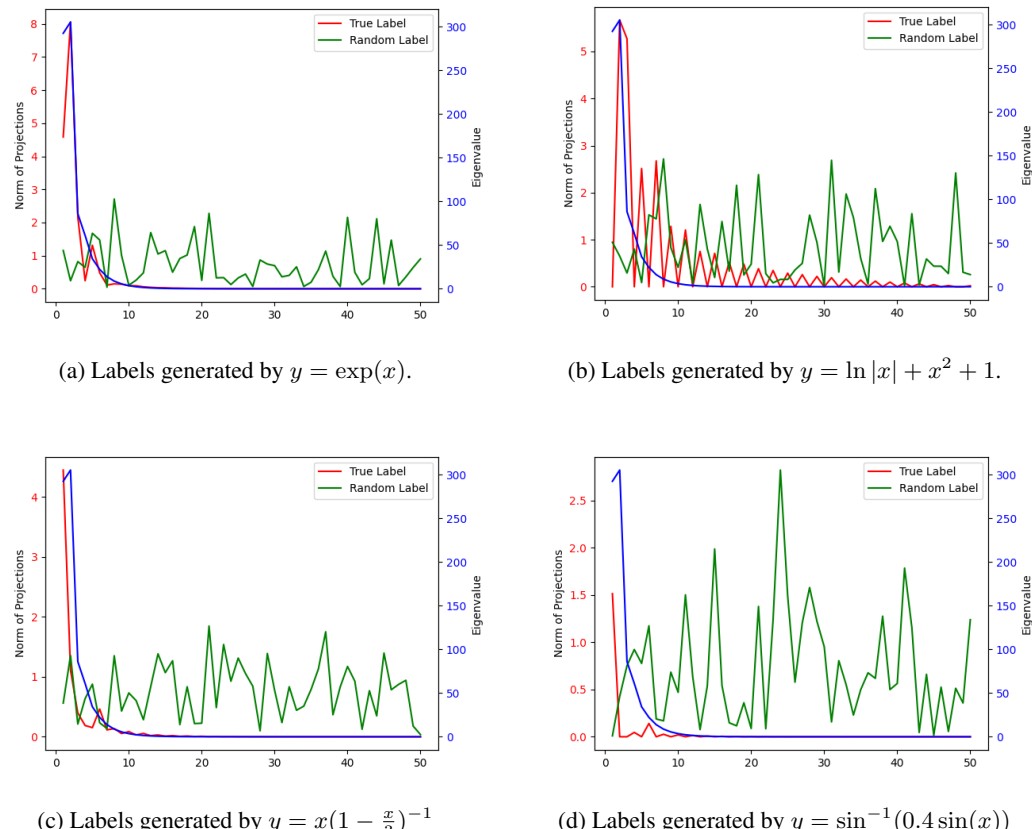

(a) Labels generated by $y = \exp(x)$.

(b) Labels generated by $y = \ln|x| + x^2 + 1$.

(c) Labels generated by $y = x(1 - \frac{x}{3})^{-1}$

(d) Labels generated by $y = \sin^{-1}(0.4\sin(x))$

Figure 8: Projections of structured label vectors onto the eigenspectrum of the KAN-TK matrix. Each plot shows how the label signal distributes across kernel eigenvectors: concentration on top eigenvalues indicates more favorable alignment and thus faster convergence.

For each function, we compute the infinite-width KAN-TK, $\boldsymbol{H}^{\infty}$, and project the corresponding label vector onto its eigenbasis.

**Results.** Figure 8 shows the projection profiles across all four functions. In every case, the structured label vectors place a substantial portion of their energy on the leading eigenvectors of the kernel, those associated with the largest eigenvalues. Such concentration indicates strong alignment with the dominant kernel directions, which in turn predicts rapid convergence under gradient descent, consistent with our theoretical characterization in Theorem 4.6. By contrast, as shown in the random-label experiments in the main text, unstructured labels distribute their energy more uniformly across the spectrum, resulting in slower and more erratic convergence.

An additional observation is that highly nonlinear mappings (e.g., $y = \exp(x)$) yield especially concentrated projections on the top eigendirections. While this may seem counterintuitive, it reflects the fact that smooth monotonic functions align well with the principal components of many kernel operators. Nevertheless, as the underlying label function becomes more intricate or oscillatory, the energy distribution spreads deeper into the spectrum, indicating reduced alignment and correspondingly slower convergence.

D.4    EXAMINING THE TIGHTNESS OF THE CONVERGENCE BOUND

To empirically assess the tightness of the linear convergence bound derived in Theorem 4.2, we compare the theoretical rate with the observed training loss. Recall that the theorem guarantees a per-iteration contraction of the loss by at least a factor of $(1 - \eta\lambda_0/2)$. We therefore plot this

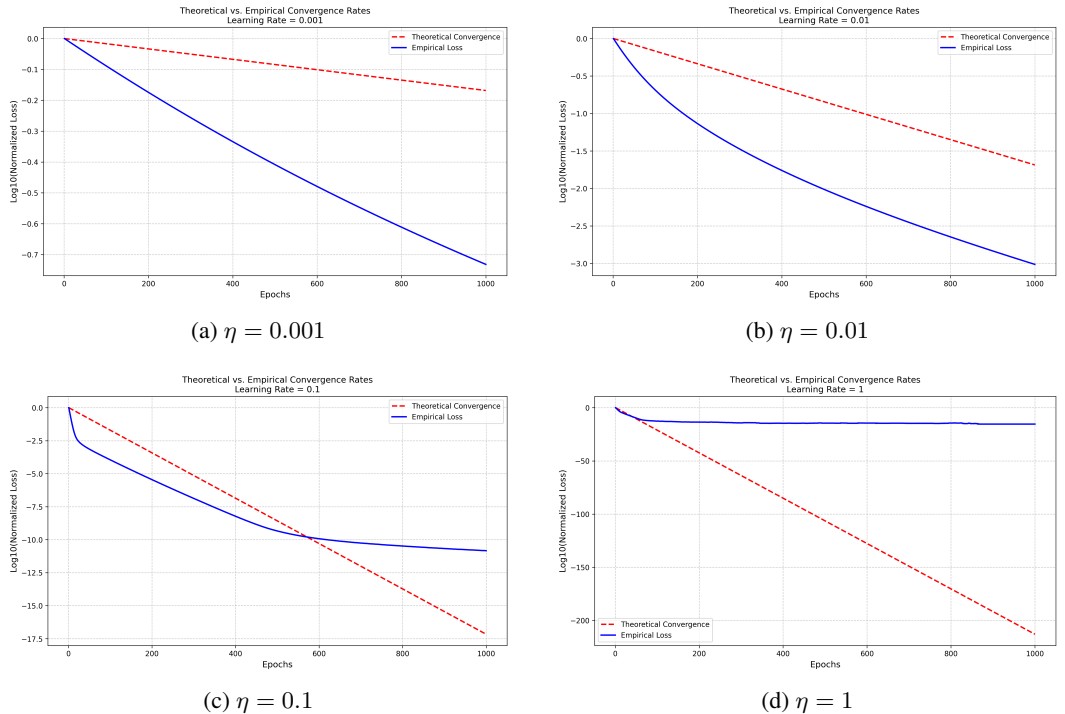

Figure 9: Comparison of the theoretical convergence bound (red dashed line) from Theorem 4.2 with the empirical training loss (blue solid line) for different learning rates.

theoretical upper bound (red dashed line) together with the empirical training loss (blue solid line) on a logarithmic scale.

**Setup.** We use a two-layer Kolmogorov-Arnold Network with hidden layer width $m = 5000$. The training set consists of $n = 10$ samples $\{(x^q, y^q)\}_{q=1}^{10}$ where the inputs $x^q$ are drawn uniformly from $[-1, 1]$, and the labels are generated according to

$$y = \exp(-x^2) + x^2.$$

Training is performed with full-batch gradient descent for 1000 epochs. Importantly, only the first-layer coefficients $\alpha_{ijk}$ are updated during training, while the second-layer coefficients $\beta_{il}$ are kept fixed, in line with the setting analyzed in Theorem 4.2. We vary the learning rate $\eta \in \{0.001, 0.01, 0.1, 1\}$.

For visualization, we report

$$\log_{10}\left(\frac{\mathcal{L}(t)}{\mathcal{L}(0)}\right),$$

that is, the base-10 logarithm of the ratio of the loss at iteration $t$ to the initial loss, and plot it against the theoretical bound $t \log_{10}\left(1 - \eta\lambda_0/2\right)$.

**Results.** As shown in Figure 9, the empirical loss decreases consistently faster than the theoretical prediction, confirming that our analysis provides a valid upper bound. The discrepancy between the empirical and theoretical curves reflects the conservatism of the bound, which is derived under worst-case assumptions. For small learning rates ($\eta = 0.001$ and $\eta = 0.01$), the loss exhibits smooth, nearly linear decay. For $\eta = 0.1$, the initial convergence is significantly faster than predicted before flattening out. For $\eta = 1$, training becomes unstable and the loss fails to decrease, in accordance with the constraints on $\eta$ imposed by the theory.

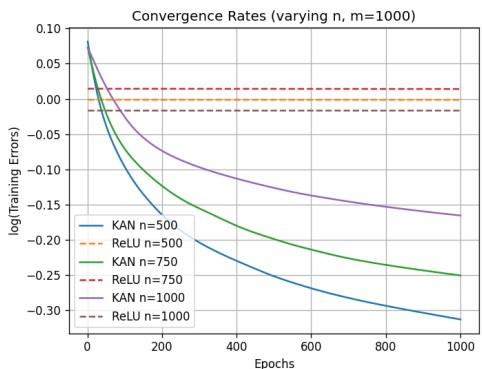

Figure 10: Convergence of FastKAN and a ReLU network with width $m = 1000$ across different sample sizes.

### D.5 KAN VS. RELU NETWORKS ACROSS DIFFERENT SAMPLE SIZES

We conducted an additional experiment to demonstrate that a one-hidden-layer KAN can outperform a one-hidden-layer ReLU network of the same width when the number of training samples is large.

**Setup.** We trained a standard neural network and a FastKAN model, each with a fixed hidden-layer width of $m = 1000$, using varying numbers of training samples $\{500, 750, 1000\}$. The experiment was performed on the synthetic dataset introduced in Section 5.1, with input dimension $d = 100$, and all models were trained for 1000 epochs.

**Results.** The results, shown in Figure 10, indicate that FastKAN exhibits substantially faster convergence compared to a ReLU network of the same width across all sample sizes.

## E RATIONALE FOR FIRST-LAYER TRAINING

In this section, we explain why we focused on first-layer training for KANs. As demonstrated in Sections E.1 and E.2, training the first layer alone outperforms training only the second layer and achieves performance comparable to full-network training. This observation supports our assumption that training only the first layer is a reasonable and efficient approach.

### E.1 COMPARISON OF FIRST AND SECOND LAYER TRAINING

Here, we compare the effects of training only the first layer versus training only the second layer.

**Setup.** We follow the same experimental setup as in Section 5.1. The dataset consists of 100 samples drawn from a 100-dimensional unit sphere, with random labels assigned to the data points. We consider network widths $m \in \{500, 1000, 2000, 4000, 8000\}$ and train for 2000 epochs. In the first experiment, the second-layer coefficients are fixed while the first layer is trainable; in the second experiment, the first-layer coefficients are fixed while the second layer is trainable.

**Results.** As shown in Figure 11, training only the first layer yields significantly faster convergence than training only the second layer, supporting the decision to focus on first-layer training.

### E.2 COMPARISON OF FIRST AND FULL LAYER TRAINING

Next, we compare training only the first layer to full-network training.

**Setup.** The experimental setup is the same as in Section 5.1, with 100 samples from a 100-dimensional unit sphere and network widths $m \in \{500, 1000, 2000, 4000, 8000\}$. Training is per-

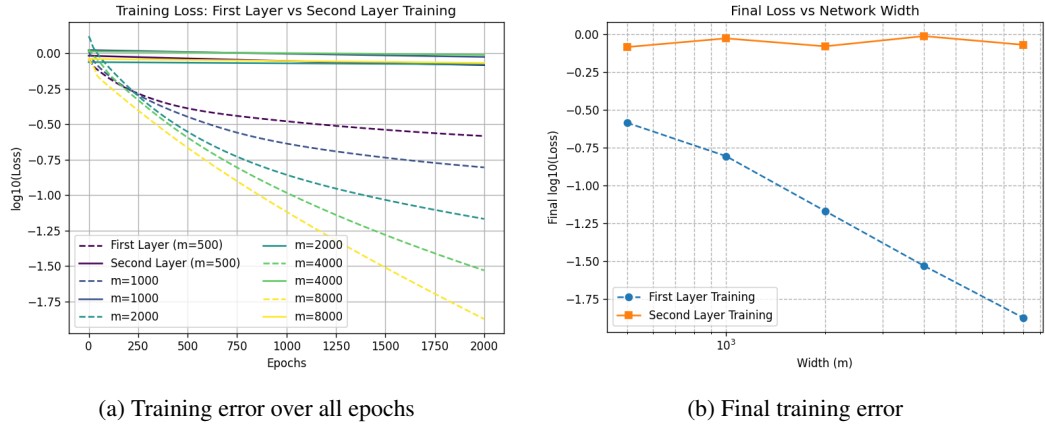

(a) Training error over all epochs      (b) Final training error

Figure 11: Comparison of first-layer and second-layer training: (a) training error over all epochs, and (b) final training error.

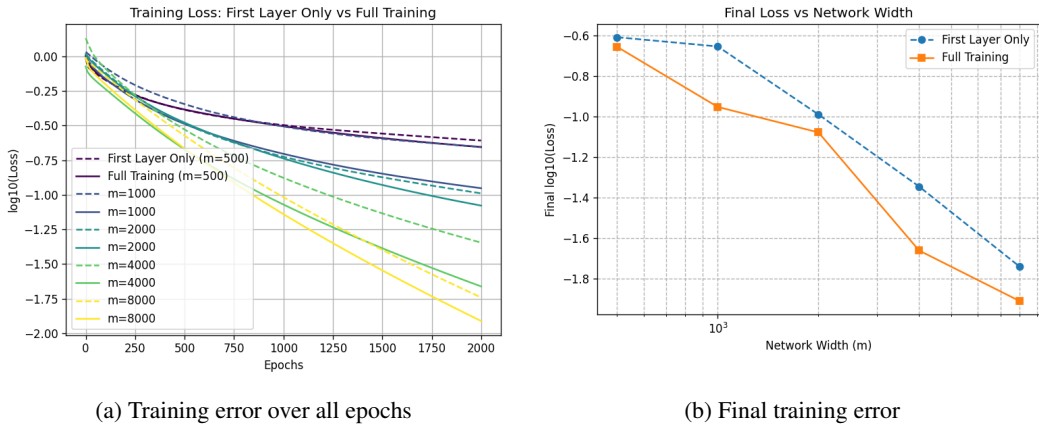

(a) Training error over all epochs      (b) Final training error

Figure 12: Comparison of first-layer and full-network training: (a) training error over all epochs, and (b) final training error.

formed for 2000 epochs. In the first experiment, only the first-layer coefficients are trained, while in the second, both layers are trained.

**Results.** Figure 12 shows that first-layer training achieves performance comparable to full-layer training. Moreover, it is more parameter-efficient and converges faster.

# F   ROLE OF LLMS IN THIS WORK

We used large language models (LLMs), including OpenAI's GPT and Google's Gemini, to assist with writing tasks and commenting on code during the preparation of this manuscript. The models were employed strictly under the direct supervision of the authors. All technical content, experiments, results, and claims in this work are entirely the responsibility of the authors, and no output from the language models was used without thorough verification.

