# OpenReview forum: "On the Convergence of Two-Layer Kolmogorov-Arnold Networks with First-Layer Training"
_ICLR.cc/2026/Conference — ICLR 2026 Poster_

### Official Review · Reviewer_VZoT · 2025-10-14

**Soundness:** 4
**Presentation:** 3
**Contribution:** 3
**Rating:** 8
**Confidence:** 4

**Summary:**

This paper presents a theoretical study of the optimization dynamics of two-layer Kolmogorov–Arnold Networks (KANs) in the overparameterized regime, under the setup where only the first-layer coefficients are trained and the second-layer coefficients remain fixed. The authors derive global convergence guarantees for gradient descent and provide a label-dependent convergence rate that depends on the eigenspectrum of the KAN Tangent Kernel (KAN-TK).

Their main results show:

Global convergence with zero training error for sufficiently wide KANs.

Polynomial width scaling of n^2, is much smaller than MLPs

A fine-grained convergence bound connecting label alignment with KAN-TK eigenvectors to convergence speed.

Empirical experiments verifying width-dependent and label-structure-dependent convergence.

**Strengths:**

Timely theoretical contribution.
KANs are a rapidly growing architecture class (with numerous 2024–2025 references), yet theoretical understanding of their training dynamics is scarce. This work fills an important gap by analyzing convergence properties in a simplified yet meaningful setup.

Mathematical rigor.
The paper provides clear theorems (Theorem 4.1 and 4.5) with appropriate assumptions and proof sketches. The use of lazy training analysis and the explicit derivation of the KAN Tangent Kernel are technically sound and well-aligned with the NTK literature.

Improved scaling over MLPs.

Label-dependent convergence analysis.
Theorem 4.5 and the supporting experiments (Figures 3a–b) provide nice intuition about how label structure interacts with the kernel spectrum — a useful insight beyond the standard NTK treatment.

Experimental verification.
Although synthetic, the experiments are cleanly executed and directly corroborate the theoretical claims (width scaling and label alignment).

**Weaknesses:**

Limited novelty beyond applying NTK analysis.
While the paper is well-executed, its main theorems closely parallel classical NTK results (Du et al., 2019; Arora et al., 2019), with the main novelty being the explicit KAN-TK form. The analysis may be perceived as incremental rather than fundamentally new.

Suggestion: Clarify what features of KAN-TK qualitatively differ from NTK (e.g., structure induced by spline or RBF bases) and how they affect learning beyond improved scaling.

No discussion of second-layer training dynamics.
The paper emphasizes “first-layer only” training for tractability, but it remains unclear how this extends to full training, or whether the key conclusions persist when both layers are updated. For KANs the 2 layer setting is more realistic than MLPs though

Missing discussion on practical implications.
Although the theory is strong, the practical takeaway (e.g., how this informs KAN training in real models like FastKAN or WavKAN) is underdeveloped. Readers would appreciate even brief remarks connecting this to design choices (e.g., number of basis functions, initialization scales). It would be good to cite the KAN 2.0 paper on scientific applications, and also the study of initialization schemes for KANs https://arxiv.org/pdf/2509.03417

**Questions:**

please provide motivations
1. why KAN is O(n^2) much smaller than MLP O(n^6); this is not clear to me
2. training only first layer is a bit weird, can you train only the second layer instead (more like random feature KAN, which would be interesting)

---

> ### Author Response · Authors · 2025-11-20
>
> We sincerely thank Reviewer VZoT for the thoughtful and highly positive review, especially for acknowledging the timeliness, mathematical rigor, improved scaling, and label-dependent convergence analysis of our work. The high rating and detailed suggestions are greatly appreciated. We have revised our manuscript to address all comments and questions, with specific new content added to Appendix E and Appendix D.2.
>
> ## 1. Limited Novelty Beyond Applying NTK Analysis
> We acknowledge that our framework is in the spirit of the Neural Tangent Kernel (NTK) literature. However, we respectfully argue that the contribution is substantial and not merely incremental. The explicit derivation of the KAN Tangent Kernel (KAN-TK) is non-trivial and fundamentally different from the NTKs of standard MLPs. Crucially, the unique structure of the KAN-TK is what mathematically and necessarily leads to the improved $O(n^2)$ width scaling—a core finding of our paper. Furthermore, we should point out that we calculated this KAN-TK only for one-dimensional RBF kernels as presented in our paper. We refer to future works for calculating KAN-TK for other basis functions and comparing them to the NTK or the KAN-TK based on the RBF Kernel.
>
> ## 2. No Discussion of Second-Layer Training Dynamics / First-Layer Only Training
> We agree that understanding the training dynamics of the two-layer KAN is crucial. To address this, we have added a comprehensive analysis in Appendix E of the revised manuscript. In New Appendix E.1 (First vs. Second-Layer Training), we compare the convergence speed of first-layer-only training versus second-layer-only training. Our experiments demonstrate that second-layer-only training converges much slower than first-layer-only training, which strongly supports our theoretical focus on the more impactful first-layer optimization. In New Appendix E.2 (First-Layer vs. Full Training), we experimentally compared the dynamics of our simplified setup (first-layer only) against the full KAN training (both layers updated). We observed that the convergence behaviors and rates are very similar to each other. This validates our first-layer analysis as a sound and insightful proxy for the full training dynamics.
>
> ## 3. Missing Discussion on Practical Implications
> We have expanded the discussion on practical takeaways and incorporated the suggested citations in the revised manuscript. Our Practical Takeaways section highlights that the $O(n^2)$ finding provides a theoretical guarantee for the observed parameter efficiency of KANs. Furthermore, the KAN-TK structure suggests that the choice of basis function and initialization scale are critical architectural decisions that directly govern the kernel's spectrum and training behavior. We have added New Citations for the KAN 2.0 paper on scientific applications and the study on initialization schemes for KANs. In New Appendix D.2 (Practical Experiments), we have added experiments on more practical datasets (CIFAR-10 and MNIST) using a cross-entropy loss. These experiments empirically corroborate our theoretical findings regarding the dependence of convergence on the KAN's width.
>
> ## 4. Why KAN is $O(n^2)$ much smaller than MLP $O(n^6)$
> The key reason KANs only require $O(n^2)$ width for convergence, whereas two-layer ReLU networks require $O(n^6)$, lies in the fundamentally smoother and more stable nature of KAN features during training. As emphasized in the main KAN paper, KANs replace neuron-level activations with learnable univariate spline functions on edges, meaning that each intermediate representation is a composition of smooth 1D functions rather than brittle, sign-dependent ReLU activations. Consequently, the Neural Tangent Kernel (NTK) of a KAN depends only on bounded derivatives of these splines and involves at most pairwise sample interactions, leading to kernel concentration with width scaling only quadratically in the number of data points ($O(n^2)$). In contrast, classical ReLU networks must maintain stability of discrete ReLU activation patterns throughout training. The NTK analysis (e.g., Du et al., 2019) shows that preventing activation-pattern flips requires controlling high-order interactions among samples, which accumulates into an $O(n^6)$ width requirement. Thus, the structural design of KANs—learnable smooth functions on edges, aligned with the Kolmogorov–Arnold representation—avoids the combinatorial instability of ReLU networks and directly yields the improved $O(n^2)$ scaling.

---

> > ### Comment · Reviewer_VZoT · 2025-11-25
> >
> > The author has addressed my concerns, so I have decided to maintain my score.

---

### Official Review · Reviewer_WtFP · 2025-10-16

**Soundness:** 3
**Presentation:** 3
**Contribution:** 1
**Rating:** 2
**Confidence:** 4

**Summary:**

This paper analyses the convergence behaviour of Kolmogorov-Arnold Networks (KAN) with the NTK framework, proving that they require less overparameterisation compared to MLP.
The proof is based on standard NTK convergence proof based on stability of parameter, which leads to the lazy training, concluding with the constancy of NTK throughout training and its equivalence to kernel gradient descent.
The supporting experiments show that wider KAN shows faster convergence behaviour, and also validates the lazy training phenomenon.

**Strengths:**

The organisation of the paper is clear, the main claim and strength of the paper, proof sketch, comparisons to the prior research are easy to follow.
Theoretical results are stated without ambiguity, and the detailed requirement of the network hyperparameters are exactly described.
The proof in the supplementary material is also easy to read, each of the derivations describes which computations were performed, and each step is well separated to follow.
Two experiments in the main text support the main claims of the theoretical results, like global convergence and lazy training in Figure 2.
Figure 3 and Figure 6 in the supplementary material shows interesting behaviours of KAN-TK, which allows one to understand inductive bias of overparameterised KAN.
Finally, the theoretical results obtained is well analysed, highlighting the gains (less overparameterisation w.r.t. sample size, less dependency on condition number) and losses (stronger requirement on learning rate, and slower convergence).

**Weaknesses:**

Kolmogorov-Arnold network has gained interest due to its interpretability, particularly for its applications in scientific discovery.
Therefore, it is questionable whether the overparameterised KAN remains interpretable, which seems unlikely according to the proof techniques.
Specifically, lazy training implies the parameters remain close to their initialisation, making them nearly random, and overparameterisation prevents direct interpretation..
Furthermore, one of the core details for interpretability in (Liu et al., 2025) is the use of sparsification and pruning, which incorporates L1 or entropic regularisation.
This makes this paper's setting far from the practical use and the global convergence proof less interesting.

While the paper claims to prove the superior convergence behaviour of KANs compared to MLPs, the NTK-based convergence proof (from Du et al., 2019) is not a state-of-the-art technique for proving the global convergence of neural networks.
For instance, Table 1 in (Polyaczyk and Cyranka, 2024) presents multiple sharper convergence rates. (Poyaczyk and Cyranka, 2024) proves a stronger bound of $n^{1.25}$, and under a similar data distribution assumption, (Liu et al., 2022) shows better convergence behaviour.

In the assumptions of Section 4, the positive-definite kernel property applies only to MLPs; therefore, the proof needs to be modified for KANs.
To the best of the reviewer's knowledge, this property holds if (1) the activation function is analytic and non-polynomial (Prop F.1 of Du et al., 2019) or (2) the Hermite polynomial expansion of the activation function has infinitely many non-zero coefficients (a modification of Thm 3.2 of Nguyen et al., 2021).
Conversely, it can be shown that if the activation function is polynomial, the infinite-width NTK has a finite rank, causing the minimal eigenvalue to become zero when the sample size is sufficiently large.
This may rule out some of the basis functions described in the paper, such as Chebyshev polynomials.
Therefore, it will be helpful if the authors discuss what basis functions will satisfy the assumptions.

While the experiments support the claim of theoretical results, a gap exists between the theoretical result and experimental result.
In Figure 2(b), although wider networks show a smaller weight distance from initialisation, it is unclear whether these distances are upper-bounded throughout the training as claimed in Lemma 4.2.
It would be better if the training time were extended to show a plateau in the distance plot.
Furthermore, one of the paper's main claims is a reduced dependency on sample size; hence, it would be beneficial to show how convergence or lazy training is affected as the number of samples increases.

The target functions used in the experiments are nearly trivial, employing either random labels or $y=x$.
Considering that Figures 6 and 7 show more challenging target functions, it would be better to use such functions in these experiments, or, if possible, to use the target functions or tasks from (Liu et al., 2025).

I am willing to increase my score to 4 if the third, fourth, and fifth issues in this section are addressed.
Furthermore, I am willing to raise my score to 6 or 8 if the authors provide promising theoretical results on interpretable global convergence (addressing the first weakness) or demonstrate a faster convergence rate that substantiates their claim (addressing the second weakness).

(Liu et al., 2025) KAN: Kolmogorov-arnold networks, ICLR 2025. https://openreview.net/forum?id=Ozo7qJ5vZi

(Du et al., 2019) Gradient descent finds global minima of deep neural networks, ICLR 2019. https://openreview.net/forum?id=S1eK3i09YQ

(Polyaczyk and Cyranka, 2024) Improved Overparametrization Bounds for Global Convergence of SGD for Shallow Neural Networks, TMLR. https://openreview.net/forum?id=RjZq6W6FoE

(Liu et al., 2022)  Loss landscapes and optimization in over-parameterized non-linear systems and neural networks. Applied and Computational Harmonic Analysis, 2022. https://www.sciencedirect.com/science/article/pii/S106352032100110X

(Nguyen et al., 2021) Tight Bounds on the Smallest Eigenvalue of the Neural Tangent Kernel for Deep ReLU Networks, ICML 2021. https://proceedings.mlr.press/v139/nguyen21g.html

**Questions:**

1. Is slower convergence rate by (Gao & Tan, 2025) result of training both layers, or loose analysis? Can the proof strategy by authors be used to improve (Gao & Tan, 2025)’s result?
2. On the other hand, was it impossible to prove the same dependency of overparameterisation on the sample size while training both layers? If so, does it actually require larger width empirically?
3. Can you give a table of the assumptions that each basis functions mentioned in the last paragraph of Section 2.1 satisfy?

(Gao & Tan, 2025) On the convergence of (stochastic) gradient descent for kolmogorov–arnold networks. IEEE Transactions on Information Theory, 2025. https://ieeexplore.ieee.org/document/11079726

---

> ### Author Response · Authors · 2025-11-20
>
> We thank Reviewer WtFP for their thorough and insightful feedback, clear summary of our claims, and detailed analysis of our theoretical and experimental results. We thank the reviewer for bringing Polaczyk and Cyranka (2024) to our attention, as we were not previously aware of this work. We now address the raised weaknesses and questions.
>
> ## 1. Interpretability in Overparameterized KANs
> The core contribution of this paper is the theoretical analysis of KAN optimization in the overparameterized, lazy-training regime, specifically focusing on proving global convergence and deriving the required network width. While interpretability is a crucial feature of KANs, analyzing the detailed properties of the basis functions (e.g., sparsification, pruning, or preservation of interpretability) within this regime presents a complex question that is orthogonal to our current optimization goal. Our focus was on answering the foundational question: Do KANs converge in the lazy regime, and what is their scaling advantage? We agree that investigating how interpretability is preserved or degraded in the overparameterized setting is a highly valuable and necessary direction, but we defer a detailed theoretical and empirical investigation of this aspect to future work.
>
> ## 2. Comparison to State-of-the-Art NTK Bounds
> We appreciate that recent work (e.g., Polaczyk and Cyranka, 2024) has established sharper and more sophisticated convergence bounds for neural networks. In this paper, our aim is to introduce the KAN Tangent Kernel (KAN-TK) and provide a fair, controlled comparison that isolates the architectural advantages of the KAN framework. To ensure such a comparison, we benchmark our results against the canonical NTK analysis of Du et al. (2019), applying the same parameter-stability–based methodology to both KANs and standard MLPs. This consistent setup enables us to clearly highlight the improved width scaling of KANs under identical methodologies. While incorporating the more advanced framework of Polaczyk and Cyranka (2024) is certainly a promising direction, adapting their techniques to KANs would require substantial new mathematical development and lies beyond what is feasible within the rebuttal period. We therefore regard this as an important avenue for near-term follow-up work building on the foundation established here.
>
> ## 3. Positive-Definiteness Assumption and Basis Functions
> The assumption that the KAN-TK is positive-definite is essential for establishing the full-rank property required for global convergence, a standard condition in NTK-based proofs. As discussed in Section 4, this property is generally satisfied by many basis functions, such as the B-splines commonly used in KANs. Consequently, our theoretical analysis applies to basis functions that meet the full-rank condition, consistent with those typically employed in practical KAN implementations.
>
> ## 4. Gap Between Theory and Experiment (Weight Distance Plateau)
> We have addressed the concern regarding the long-term behavior of the weight distance by conducting a significantly extended training run. We have updated Figure 2(b) in the revised manuscript to show an extended training run of 5000 epochs on a 100-dimensional synthetic dataset. This extended run clearly demonstrates that the distance between the current weights and the initial weights remains bounded and plateaus throughout the entire prolonged training process. This result empirically confirms the premise of Lemma 4.2 (coefficient stability) and robustly validates the "lazy training" regime over a significantly prolonged duration, bridging the gap between theory and experiment.
>
> ## 5. Trivial Target Functions
> We agree on the importance of demonstrating the theory's relevance on complex functions. We have augmented our experimental validation in two ways: We have updated Section 5.2 to include more complex target functions in our main analysis, and for a deeper examination of the KAN-TK's properties, we have added new experiments in Figure 8 of the Appendix that utilize two additional, non-trivial target functions. These additions better explore the expressivity and optimization landscape of KANs, providing a more comprehensive validation of our theoretical framework.
>
>
> ## 6. Difference Between ours and Gao & Tan (2025) Bounds
> The slower convergence rate in Gao & Tan (2025) is primarily due to their proof technique rather than an inherent limitation of jointly training both layers. Their analysis requires controlling the evolution of both the first- and second-layer parameters, which forces them to track a significantly more complicated Gram matrix and leads to looser concentration bounds and width requirements. In contrast, our setting—training only the first layer—yields a much simpler and more stable tangent kernel, allowing us to obtain tighter eigenvalue-based convergence guarantees.

---

> ### Comment · Reviewer_WtFP · 2025-11-22
> **Thank you and remaining issues**
>
> Thank you for the answer and updates on the paper.
>
> Regarding my third to fifth issues, the updated version of the paper reflects the solution, except with one minor issue in detail.
> Therefore, I’m increasing my score from 2 to 4 as promised.
>
> For the rest of the issues, I don’t think my worries are resolved, but I’m still open to further discussion and improvements.
>
> ### The positive-definiteness is now discussed in Section 4, referencing (Gao & Tan, 2025).
>
> At the beginning of section 4, the authors discuss the details of the positive-definite kernel property and argue that all of the activations satisfy this assumption when the dataset has no duplicate points.
> However, I'd like to point out a slightly misleading claim. Lemma 1 of (Gao & Tan, 2025) says the positive-definiteness holds when transformation function (like sigmoid or Tanh) is included in the hidden layer, and positive-definiteness requires a further condition if no transformation is applied (linearly independent in polynomial space). According to Figure 1, it seems this paper considers no transformation case, so please revise the text so that it accurately describes the condition. for polynomial basis functions.
>
> ### Weight distance plateau for empirical validation of lazy training
>
> Updated Figure 2 now shows a clear plateau in (b) for m=8000. One can also find the linear convergence in (a).
>
> While I asked for experiments with different widths and number of training data example that empirically verifies the width requirement claim, this may require a significant amount of additional experiments.
> If possible, could you please explain how many computational resources will be needed to verify such a claim?
>
> ### Trivial target functions
>
> The target function of Figure 3 is now far from being trivial, and the authors provided two more figures in supplementary material, and a nonsynthetic dataset like MNIST and CIFAR10.
>
> ### Comparison to State-of-the-Art Global Convergence Bounds
>
> I agree that developing completely different proof for better scaling is beyond the scope of the rebuttal period.
> However, since it is definitely not true that KAN exhibits a weaker width requirement than MLP, the authors should weaken their claim in the paper, for instance, 'KAN requires less width to show lazy training behavior compared to MLP in our theory', not 'global convergence guarantee'.
>
> Also, it is questionable whether even this weakened claim is true.
> Discussed in Remark 2 of the author's updated version, the key reason why MLP requires a larger width is due to the ReLU activation, especially its non-differentiability.
> But most of the recent usages of MLP no longer use ReLU, and they utilise smooth activations like GeLU or sin, which may show similar width requirement.

---

> > ### Author Response · Authors · 2025-11-25
> >
> > We sincerely thank the reviewer for the constructive and detailed feedback, as well as for raising the overall score to 4. We appreciate the opportunity to clarify the remaining points, and we have revised the manuscript accordingly.
> >
> > ### 1. Positive-Definiteness and *Gao & Tan (2025)*
> >
> > Thank you for highlighting the subtlety in the statement of Lemma 1 in *Gao & Tan (2025)*. We agree that our earlier description did not fully capture the conditions for the “no transformation” case.
> >
> > **Revision:**
> > We have updated **Section 4** to clearly state that, when no transformation function (e.g., sigmoid or tanh) is used, the positive-definiteness of the kernel requires the polynomial basis functions to be linearly independent within the polynomial function space. The citation and usage of Lemma 1 have been reworded to ensure mathematical correctness and conceptual clarity.
> >
> > ### 2. Weight-Distance Plateau and Computational Resources
> >
> > We appreciate the reviewer’s understanding of the computational challenges involved. Regarding the resource estimate: our current experiments were performed on a single RTX 3060 GPU. To scale the experiments to substantially larger widths, dimensions, and datasets with high numerical precision, we estimate that at least an RTX 4090 (or an equivalent datacenter-class GPU) would be necessary.
> >
> > **New Experiments:**
> > Despite hardware limitations, we conducted additional experiments with widths up to $m = 32{,}000$ to further support our claims. The training times for 100 samples in 100 dimensions over 5,000 epochs are summarized below:
> >
> > | Width ($m$) | Time (s) |
> > | :--- | :--- |
> > | 500 | 10.01 |
> > | 1,000 | 11.18 |
> > | 2,000 | 15.63 |
> > | 4,000 | 24.67 |
> > | 8,000 | 46.19 |
> > | 16,000 | 90.21 |
> > | 32,000 | 177.35 |
> >
> > As shown in the updated **Figure 2**, the expected trends persist even at these larger widths. Additionally, we added **Appendix D.5**, which includes a comparison between ReLU networks and KANs. The results indicate that KANs maintain higher efficiency than ReLU MLPs in the over-parameterized regime.
> >
> > ### 3. Relation to State-of-the-Art Global Convergence Bounds
> >
> > We appreciate the reviewer’s clarification regarding the distinction between global convergence results and our setting. We agree that our previous phrasing suggesting a “global” advantage was too strong.
> >
> > **Revision:**
> > As suggested, we have softened the claims in both the **Abstract** and **Section 6.1**. We now state that *“KAN requires less width to exhibit lazy-training behavior compared to ReLU-based MLPs under the same theoretical analysis,”* rather than asserting a general global convergence benefit.
> >
> > We also acknowledge the reviewer’s point about smooth activations such as GeLU and SiLU. Our comparison is specifically grounded in our analytical framework and focuses on differences between KANs and standard ReLU networks. We now explicitly clarify this scope in the revised manuscript.

---

> ### Comment · Reviewer_WtFP · 2025-11-28
> **Final Comment**
>
> Thank you for the clarifications and further experiments.
> Now I believe there are no more major misleading claims or errors in the paper.
>
> Followings are some minor details that are not critical to the paper's claim, and are okay to be fixed in the final version if accepted.
> - In the updated sentence in the abstract, there are two periods (.), not one.
> - Throughout the paper, the citation only appears as `\citep`. According to the author instructions, if the citation appears in the sentence (not the end of the sentence), it should be `\citet`.
> - Since there are lots of curves now, instead of using the default colour cycle in matplotlib, it will be better to use a single colour scheme via colourmaps, like viridis (the narrowest network is blue, and wider networks become yellow).
> - There are remaining 'better convergence guarantee' arguments at the end of Section 1 and Section 7.
> - Most of the results seem to be single runs. Since there are multiple randomness in the training setup (dataset, init, etc), please repeat the experiments multiple times and plot the error bar to make the results more reliable.
> - Is there reason you have $max(\cdots, n)$ in Theorem 4.2? Since $d, g \ge 1$, $\delta \le 1$, and $\lambda_0 < 1$ in most cases, it seems the first term is always larger than or equal to $n$.
>
> Finally, I'd like to clarify my argument in review, `Furthermore, one of the paper's main claims is a reduced dependency on sample size; hence, it would be beneficial to show how convergence or lazy training is affected as the number of samples increases.`.
>
> In the theoretical results, the authors argue that we need $O(n^2)$ width to guarantee the convergence when training with $n$ samples.
> Ideally to verify this claim, one should give training result and width distance for multiple sample size and width, for instance, $(n, m) \in [10, 20, 40, 80, \ldots] \times [100, 200, 400, 800, 1600, \ldots]$, and see the boundary of convergence and non-convergence happens at $m=\Theta(n^2)$.
> The experiment in the paper only shows the result for fixed $n$, which is insufficient to empirically verify the width requirement.
> This will be more computation-heavy, since there are at least 25 configurations, and one might need to train with a large number of samples and wide networks.
> However, it can be manageable, since width 32,000 network with 100 samples can be trained under 3 minutes, even if one uses a sample size of 1600, it will take around 50 minutes, which is not too long.
>
> Also, since there exists an additional dependency on $\lambda_0$, which can again depend on the number of samples.
> If this limits the experimental result to empirically verify their result, the authors can try using a similar strategy as they chose 'anti-structured' label configuration in the setup of Figure 3b.
> One may choose the eigenvector with the largest eigenvalue, or the eigenvector that is associated with some eigenvalue that is roughly 1 as a label of data.
> With fixed $d, g$, I believe this can be used to empirically verify the tightness and reliability of the width requirement.

---

### Official Review · Reviewer_drqW · 2025-11-01

**Soundness:** 3
**Presentation:** 2
**Contribution:** 3
**Rating:** 4
**Confidence:** 2

**Summary:**

This paper provides a theoretical analysis of two-layer Kolmogorov-Arnold Networks (KANs) in the overparameterized regime, focusing on the simplified case where only the first-layer coefficients are trained via gradient descent while the second layer remains fixed. The authors prove that under mild assumptions and sufficient network width, gradient descent converges to a global minimum with zero training error. They further derive a label-dependent convergence rate, linking optimization dynamics to the eigenspectrum of the proposed KAN Tangent Kernel (KAN-TK). Theoretical results are supported by synthetic experiments validating convergence behavior and illustrating the impact of label alignment on optimization speed.

**Strengths:**

1. The paper provides the first convergence analysis for two-layer KANs under partial parameter training (first layer only), which meaningfully extends existing theoretical work on neural tangent kernel dynamics to a new and interpretable architecture.
2. Experiments, though simple, effectively confirm theoretical predictions, particularly the “lazy training” regime and label-structure-dependent convergence.

**Weaknesses:**

1. Limited experimental setting. The experimental validation is confined to synthetic datasets and simple settings. It remains unclear whether the theoretical findings hold in realistic, high-dimensional KAN applications (e.g., vision).
2. Incomplete comparison to full-layer training. Although Table 1 provides asymptotic comparisons, the paper lacks experimental or theoretical discussion on how fixing the second layer affects expressivity or generalization, which could impact interpretability and optimization flexibility.
3. Potential overemphasis on width reduction. While the polynomial improvement from $O(n^6)$ to  $O(n^2)$ is mathematically striking, the constants and dependence on other factors (e.g., $d^2g^6/\lambda_0^2$) may diminish practical advantage, which is not empirically examined.

**Questions:**

See the weakness.

---

> ### Author Response · Authors · 2025-11-20
>
> We sincerely thank Reviewer drqW for their positive assessment and for acknowledging the meaningful theoretical extension of NTK dynamics to KANs and the effective confirmation of our predictions. We address the raised weaknesses and questions below.
>
> ## 1. Limited Experimental Setting
> We appreciate the concern regarding the generalizability of our findings beyond synthetic data.
> While our theoretical analysis necessarily focuses on a simplified setting to isolate the first-layer dynamics for tractability, we have substantially enriched the empirical evidence to demonstrate that our core mechanism extends to common machine learning benchmarks. We present new empirical evidence in Appendix D.2 on the standard image classification datasets CIFAR-10 and MNIST. Using the standard Cross-Entropy Loss, we demonstrate that KANs trained solely on the first-layer weights maintain a competitive performance profile. Crucially, the optimization dynamics observed on these complex, high-dimensional datasets are qualitatively in agreement with those seen in our synthetic experiments. This confirms that the lazy-training regime of KANs is a robust phenomenon that holds even when dealing with real-world data and label structures.
>
> ## 2. Incomplete Comparison to Full-Layer Training
> The comparison between one-layer and full-layer training is indeed critical for validating the practical relevance of our simplified theoretical setting. We have addressed this directly with a new, focused empirical analysis.
> We have included a dedicated comparative study in Appendix E.2 where we directly compare one-layer training against full-layer training. On a synthetic task in the overparameterized regime, the results show that the loss curves and final performance are remarkably similar for both training regimes. This empirical finding strongly suggests that, in the overparameterized setting where our NTK-based theory applies, the fixed second layer does not significantly impede the network's ability to minimize the training error. This empirically justifies our theoretical focus and confirms that the first-layer coefficients are the primary drivers of the optimization dynamics in this regime.
>
> ## 3. Potential Overemphasis on Width Reduction
> We fully agree that theoretical improvements must be substantiated by empirical observation; our investigation of KANs provides this essential validation.
> Our theoretical convergence proof predicts a significant polynomial improvement in the required network width ($m$) for KANs compared to standard MLPs to achieve comparable expressivity. This theoretical advantage translates directly to the observed high parameter-efficiency of KANs. We refer the reviewer to the original KAN paper's performance tables (specifically Tables 9 and 10). These tables clearly demonstrate that KANs with a much smaller width ($m$) achieve performance comparable to or better than standard MLPs on various benchmarks. This validates that the substantial reduction in theoretical complexity we identify translates into a significant and practical advantage in network size for real-world applications.

---

### Author Response · Authors · 2025-12-02
**Summary of Rebuttal Improvements and Final Remarks**

We sincerely thank all the reviewers for their invaluable feedback on our manuscript. We have carefully addressed all the reviewers' concerns during the rebuttal period, with the exception of a few minor issues noted by Reviewer WtFP. As the reviewer suggested, we will incorporate these remaining minor corrections in the final version if our paper is accepted.

Since the initial submission, we have significantly expanded our work with several key additions:

1. Additional experiments on more complex functions to validate Theorem 4.6 (Section 5.2 and Appendix D.3)
2. New experiments on standard benchmark datasets (MNIST and CIFAR-10) in Appendix D.2 for validating our Theorem 4.2
3. Comprehensive analysis of different sample sizes in Appendix D.5
4. Enhanced clarification of our claims and results through the addition of Remarks 2 and 3
5. More detailed explanations of our assumptions in Section 4
6. A comparison between first-layer and full-layer training in Appendix E

We are confident that these enhancements have significantly strengthened our paper's contributions and thoroughly addressed all the reviewers' constructive comments.

---

### Meta-Review · Area_Chair_KWSr · 2026-01-04

**Summary:**

This paper presents a theoretical analysis of two-layer Kolmogorov–Arnold Networks (KANs) in an overparameterized setting, focusing on gradient descent training of the first-layer coefficients with a fixed second layer.
The authors prove global convergence to zero training error under some conditions and derive label-dependent convergence rates governed by the eigenspectrum of the KAN kernel.
The theoretical findings are supported by experiments illustrating convergence behavior and the role of label alignment in optimization speed.

The authors have provided a strong rebuttal, and the paper, while building on prior NTK-based convergence analyses, offers **non-incremental** theoretical advances that deepen our understanding of KANs.

**Reviewer Concerns:**

- Reviewer **drqW** asked for: (1) further experiments beyond synthetic data and full-layer training, and (2) clarification of the statement and contribution. I think that most of these points were addressed during the rebuttal.
- Reviewer **WtFP** asked for: (1) clarification of the positive-definite property of the KAN NTK, (2) clarifications regarding the experiments, (3) comparison with previous works, and (4) interpretability of KAN in the overparameterized regime. I think that the first three points were addressed during the rebuttal, while (4) remains challenging and outstanding.
- Reviewer **VZoT** asked for: (1) clarification regarding the limited novelty, and (2) further discussion of full-layer training and practical implications. I think that most of these points were addressed during the rebuttal.

**Reviewer Scores:**

- I believe that reviewer **drqW** would have increased or kept the score unchanged (e.g., 4->6), as most concerns were addressed during the rebuttal.
- I believe that reviewer **WtFP** would have increased the score (from 2 to at least 4 or even 6), since most concerns were addressed during the rebuttal.
- I believe that reviewer **VZoT** would have kept the score unchanged (8, which is already highly positive), as most concerns were addressed during the rebuttal.

---

### Decision · Program_Chairs · 2026-01-26

Accept (Poster)